# A modular framework for multiscale, multicellular, spatiotemporal modeling of acute primary viral infection and immune response in epithelial tissues and its application to drug therapy timing and effectiveness

T. J. Sego[1,2☯]*, Josua O. Aponte-Serrano[1,2☯], Juliano Ferrari Gianlupi[1,2], Samuel R. Heaps[1], Kira Breithaupt[1,3], Lutz Brusch[4], Jessica Crawshaw[5], James M. Osborne[5], Ellen M. Quardokus[1], Richard K. Plemper[6], James A. Glazier[1,2]

**1** Department of Intelligent Systems Engineering, Indiana University, Bloomington, Indiana, United States of America, **2** Biocomplexity Institute, Indiana University, Bloomington, Indiana, United States of America, **3** Cognitive Science Program, Indiana University, Bloomington, Indiana, United States of America, **4** Center for Information Services and High Performance Computing (ZIH), Technische Universität, Dresden, Germany, **5** School of Mathematics and Statistics, University of Melbourne, Melbourne, Australia, **6** Institute for Biomedical Sciences, Georgia State University, Atlanta, Georgia, United States of America

☯ These authors contributed equally to this work.
* tjsego@iu.edu

## Abstract

Simulations of tissue-specific effects of primary acute viral infections like COVID-19 are essential for understanding disease outcomes and optimizing therapies. Such simulations need to support continuous updating in response to rapid advances in understanding of infection mechanisms, and parallel development of components by multiple groups. We present an open-source platform for multiscale spatiotemporal simulation of an epithelial tissue, viral infection, cellular immune response and tissue damage, specifically designed to be modular and extensible to support continuous updating and parallel development. The base simulation of a simplified patch of epithelial tissue and immune response exhibits distinct patterns of infection dynamics from widespread infection, to recurrence, to clearance. Slower viral internalization and faster immune-cell recruitment slow infection and promote containment. Because antiviral drugs can have side effects and show reduced clinical effectiveness when given later during infection, we studied the effects on progression of treatment potency and time-of-first treatment after infection. In simulations, even a low potency therapy with a drug which reduces the replication rate of viral RNA greatly decreases the total tissue damage and virus burden when given near the beginning of infection. Many combinations of dosage and treatment time lead to stochastic outcomes, with some simulation replicas showing clearance or control (treatment success), while others show rapid infection of all epithelial cells (treatment failure). Thus, while a high potency therapy usually is less effective when given later, treatments at late times are occasionally effective. We illustrate

**Data Availability Statement:** All software are available in GitHub repository: github.com/covid-tissue-models/covid-tissue-response-models.

**Funding:** JAG, TJS, JFG and JAS acknowledge funding from National Institutes of Health grants U24 EB028887 and R01 GM122424 and National Science Foundation grant NSF 1720625. LB acknowledges grant 01ZX1707A within the German e:Med initiative (BMBF) and grant 391070520 from the German Research Foundation (DFG). R.K.P. was supported, in part, by National Institutes of Health grants R01 AI071002, R01 AI141222 and R01 HD079327. This research was supported in part by Lilly Endowment, Inc., through its support for the Indiana University Pervasive Technology Institute. The funders had no role in manuscript preparation or the decision to submit the work for publication.

**Competing interests:** I have read the journal's policy and the authors of this manuscript have the following competing interests: JAG is the owner/operator of Virtual Tissues for Health, LLC, which develops applications of multiscale tissue models in medical applications and is a shareholder in Gilead Life Sciences.

how to extend the platform to model specific virus types (*e.g.*, hepatitis C) and add additional cellular mechanisms (tissue recovery and variable cell susceptibility to infection), using our software modules and publicly-available software repository.

## Author summary

This study presents an open-source, extensible, multiscale platform for simulating viral immune interactions in epithelial tissues, which enables the rapid development and deployment of sophisticated models of viruses, infection mechanisms and tissue types. The model is used to investigate how potential treatments influence disease progression. Simulation results suggest that drugs which interfere with virus replication (e.g., remdesivir) yield substantially better infection outcomes when administered prophylactically even at very low doses than when used at high doses as treatment for an infection that has already begun.

## Introduction

The current global pandemic of COVID-19, caused by the novel coronavirus Severe Acute Respiratory Syndrome Coronavirus 2 (SARS-CoV-2), has motivated the study of beta corona-virus diseases at multiple spatial and temporal computational modeling scales [1]. The time course, severity of symptoms and complications from SARS-CoV-2 infection are highly variable from patient to patient [2]. Mathematical modeling methods integrate the available host- and pathogen-level data on disease dynamics that are required to understand the complex biology of infection and immune response to optimize therapeutic interventions [3–5]. Mathematical models and computer simulations built on spatial and ODE frameworks have been extensively used to study in-host progression of viral infection [6], with a recent acceleration in the development of spatial COVID-19 viral infection models in response to the global pandemic [7,8].

Building multiscale models of acute primary viral infection requires integration of submodels of multiple biological components across scales (*e.g.*, viral replication and internalization, immune system responses). Non-spatial, coupled ordinary differential equation (*ODE*) models can represent many aspects of pathogen-host interaction. In the context of viral infection dynamics, specialized ODE models can describe both the entire virus-host response at the tissue and organ levels and different stages of the viral replication cycle within cells, such as binding and internalization [9,10], viral genome replication and translation [11,12], assembly, packaging and release [13,14]. By fitting ODE models to clinical or experimental data, researchers have been able to estimate important parameters, such as the turnover rate of target cells, average lifetimes of viral particles and infected cells and the rate of production of new viral particles by infected cells [15]. Other ODE models include pharmacokinetic models of drug availability [16].

The simplest non-spatial models assume that the distribution of the modelled quantities (*e.g.*, cells, viruses, chemical species) are uniformly distributed in space and time [17]. This assumption might not be realistic in solid tissues, where viruses and host immune cells are not usually distributed homogeneously and infection propagates locally [15] or in situations where transport limits the dynamics (*e.g.* the migration of antigen presenting cells to lymph nodes, the transmission of virus between organs or the microdosimetry of a drug therapy). By

averaging over spatio-temporal and individual cell variations, non-spatial models may not accurately reflect the effects of tissue heterogeneity and resulting viral infection dynamics [18]. Compartmental ODE models, like physiologically based pharmacokinetic models (PBPK) models or multi-compartment tissue infection models, maintain some of the simplicity of single-compartment ODE models, while recognizing the critical role transport can play in viral infection, immune response and treatment [19].

However, the spread of, and response to, some viruses is highly spatially localized, both *in vitro* and *in vivo* [20,21]. For example, COVID-19 often begins with infection localized to the nose and throat and then spreads to the lungs [22], with the specific location, size and distribution of lesions affecting clinical outcomes [23]. Spatial models have been increasingly used to address such issues [24], including partial differential equations [25,26] fluid-dynamic models [27] and agent-based models (ABM) [28]. ABMs represent host cells as spatially located, individual agents, and propagation of the infection emerges from individual interactions between agents. ABMs are also well suited for extending existing models by modular integration of biological subcomponents, and their model parameters should be validated by experiment and studied through sensitivity analysis [17]. ABMs have been developed to account for infection dynamics in different biological compartments (such as the lung and lymph nodes [29,30]) and to model disease progression of HIV [15,25,31–33] and dissemination of influenza virus to the lower respiratory tract [18,34].

Spatial models often predict significantly different viral and immune dynamics, parameter estimates and therapy efficacies from their non-spatial counterparts. Stochastic effects arising from spatial conditions, such as local availability of target and immune cells, greatly influence establishment of infection and lead to different infection outcomes [15]. Non-spatial models generally produce viral load titers higher than spatial models, and the peaks of infection happen significantly earlier [17,18]. Homogenous recruitment of immune cells in spatial models matches ODE models when the number of infected is large, but not at the beginning of the infection when the number of infected cells is small [18]. These differences can lead to inaccurate estimates of important parameters such as viral infectivity [35], viral diffusion [17] and the basic reproductive ratio [35]. The basic reproductive ratio is clinically needed to determine therapeutic effectiveness [35]. Microdosimetry is another area where spatial modeling is important, since spatial variation in bioavailability can lead to low concentrations in some regions of an infected tissue, which can promote the evolution of resistant viral strains [15].

In this paper, we consider *primary* infection, that is, infection by a virus which the immune system has not previously encountered, so that there is no initial adaptive immune response. We focus on *acute* cases (cases with relatively rapid onset and short duration), in which a properly functioning immune system eventually eliminates the virus completely (*clearance*). Here we review relevant components of the immune system. Some of these components are included in this work and some are not modeled. However, by constructing a modular, extensible modeling framework and computational implementation, we enable the modeling of all these components.

While the details of infection vary by virus and patient [36], infection generally begins when a virus breaches the barrier of one or more tissues causing a limited number of target cells to be exposed and then internalize the virus (Fig 1). The virus begins to replicate within the initially infected target cells, but cells do not release any newly synthesized virus for a period of hours to days (the *eclipse* or *lag* phase of infection). Within hours, infected cells release proinflammatory cytokines and complement proteins as warning signals to neighboring cells [37,38]. Some of these cytokines, like Type 1 interferons, can induce autocrine and paracrine anti-viral responses (*e.g.*, inhibiting viral replication, viral entry or inducing cell death) [39]. Cytokines recruit circulating immune cells from the blood to the infected tissue

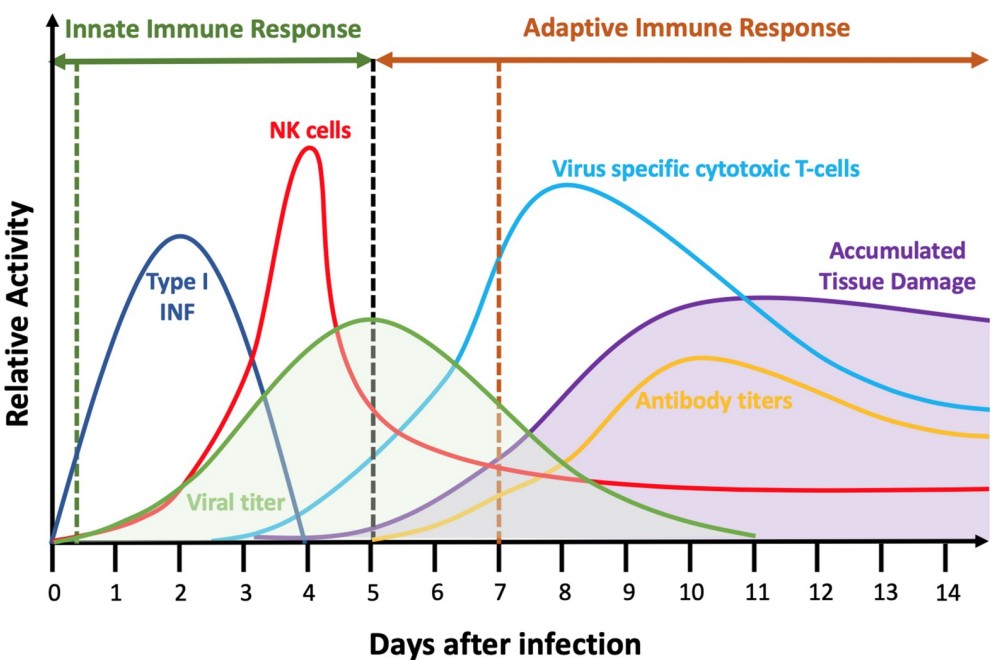

**Fig 1. Schematic of the innate and adaptive immune response during primary acute viral infection.** Exposure to the virus occurs at time 0 and extracellular viral load begins to rise (shaded green curve). Initial innate immune responses include phagocytosis of virus by neutrophils and macrophages, Type I interferon-induced antiviral resistance (*IFN*) (dark blue) and killing of infected cells by Natural Killer (NK) cells and other cell types (red). The black vertical dashed line denotes the transition between innate and adaptive immune responses. The adaptive immune response is triggered both by cytokine signaling to the lymph nodes and the migration of antigen-presenting cells from the tissue to the lymph nodes (not shown). In the later phases of infection, innate immune responses continue, but additional adaptive immune components come into play, including virus-specific cytotoxic T-cells (light blue) kill infected cells directly and also kill nearby cells through a variety of mechanisms. The orange vertical dashed line denotes the onset of the humoral adaptive immune response. B-cells produce virus-specific antibodies (orange line) which bind and inactivate virus directly and also allow its clearance and clearance of infected cells by other cell types. Tissue damage (shaded purple curve) accumulates due to cell death from direct responses to virus and from immune-cell killing by contact-mediated, diffusible factor-mediated and bystander-mediated mechanisms and eventually dissipates as cells proliferate to repair the damage (Adapted from [50,51]). The specific time course of all components varies among viruses, host tissues and host species, but the general sequence of events and immune response components are generally preserved.

and attract immune cells within the tissue by chemotaxis [40–42]. The early innate immune response activates a number of cell types including dendritic cells, macrophages, neutrophils, mast cells, basophils, eosinophils, leukocytes, and natural killer (*NK*) cells [43]. Many of these immune cells themselves release both pro- and anti-inflammatory signals. Immune signals also recruit circulating neutrophils in the blood and, later, activate cytotoxic innate immune cells like NK cells within the tissue, which kill infected cells through release of diffusible factors and contact-mediated interactions, respectively.

The temporal dynamics of the concentration of extracellular virus varies greatly among virus families, tissues and host species [44]. However, for many viruses, including influenza and coronaviruses, once infected cells begin to release virus, the amount of extracellular virus increases exponentially over a period of a few days, reaching a peak during an early phase of infection [45]. As the viral load increases, immune signaling increases rapidly (this increase is associated with the onset of fever and other symptoms) recruiting more circulating cells of the innate immune system to the infection site [46].

Immune signals from infected cells and innate immune cells help trigger the adaptive immune response. Macrophages and dendritic cells that have engulfed and degraded viral

pathogens or fragments of dead infected cells (*i.e.*, phagocytosis) migrate over a period of days to nearby lymph nodes and serve as viral antigen presenting cells (*APCs*) to naive T-cells. Antigen presentation induces naive T-cell proliferation and differentiation into pathogen-specific memory and effector T-cells [47]. Effector T-cells migrate to the site of infection and induce apoptosis of infected cells by antigen recognition and contact killing. Both infected and uninfected cells in contact with dying cells can die through bystander-effect mechanisms. In acute infections, adaptive immune response leads to pathogen neutralization and clearance [48]. Viral loads usually decrease rapidly as adaptive immune cells like CD8+ T-cells enter the tissue and eliminate infected cells. Cells also begin to send out anti-inflammatory signals, shutting down the immune response as viral clearance proceeds. Antigen presentation within the immune system also induces activation of naive B-cell lymphocytes into antibody-producing memory B-cells and plasma cells, which leads to the production of antibodies. The adaptive immune response remembers its exposure to previous pathogens and provides the body with pathogen-specific effector cells and antibodies which neutralize and clear them, providing long term immunity [49]. Tissue damage results from virus and cytokine-induced cell death (which is first noticeable after 2 or 3 days) and from killing of infected and uninfected cells by immune cells, which increases steadily until the end of viral clearance (7–10 days). Tissue recovery and healing start around the time of viral clearance and may last for several weeks.

In this paper we develop a framework for the multiscale multicellular spatiotemporal simulation of the complex processes of infection and immune response in a small patch of epithelial tissue. The model provides a representation of the complex biology that reproduces key observed emergent behaviors of infection dynamics. We create representations of the main types of components and biological mechanisms associated with acute, primary viral infection and immune response, with a special emphasis on modularity of mathematical forms and computational implementation to support the development of more detailed models in future work (*e.g.*, the creation of additional cell types, signals and detailed cell responses of various aspects of the immune response). We illustrate such an activity of development by integrating a detailed viral replication model for hepatitis C virus as an extension to the framework (see *Integration of an explicit RNA synthesis model allows the spatiotemporal modelling of hepatitis C virus infection*).

Our base model consists of three interconnected components (Fig 2A): an epithelium component, an extracellular environment component and a lymph node component. The model represents the epithelium as a compact monolayer of initially identical immobile (which is appropriate for an epithelium) epithelial cells that it classifies by their current state of viral infection (*i.e.*, uninfected, infected, virus releasing, dead, Fig 2C). These epithelial cells are initially identical in their number of viral receptors (though we show how to include heterogeneity in *Heterogeneous susceptibility inhibits spread of infection*). The cells internalize extracellular virus, modulate their number of surface receptors, replicate virus, release virus and die in response to virus production, and release an extracellular cytokine signal when infected. Our model omits interferon-induced antiviral resistance, which can be implemented as a model extension using mechanisms demonstrated in this work (*e.g.*, secretion of cytokine, modulation of internalization parameters, see *S2 Text, Developing a Model Extension in CompuCell3D*).

The model represents the extracellular environment as a space above the epithelium which provides the space in which immune cells are recruited and move, and into which cells release viruses and chemicals. We include a single type of immune cell that exhibits many key immune-cell behaviors associated with macrophage, neutrophil, NK cell and T-cell roles, including activation, chemotaxis, relaying and amplification of cytokine signals, contact killing, secretion of diffusible killing factors, and bystander killing, to represent the main types of

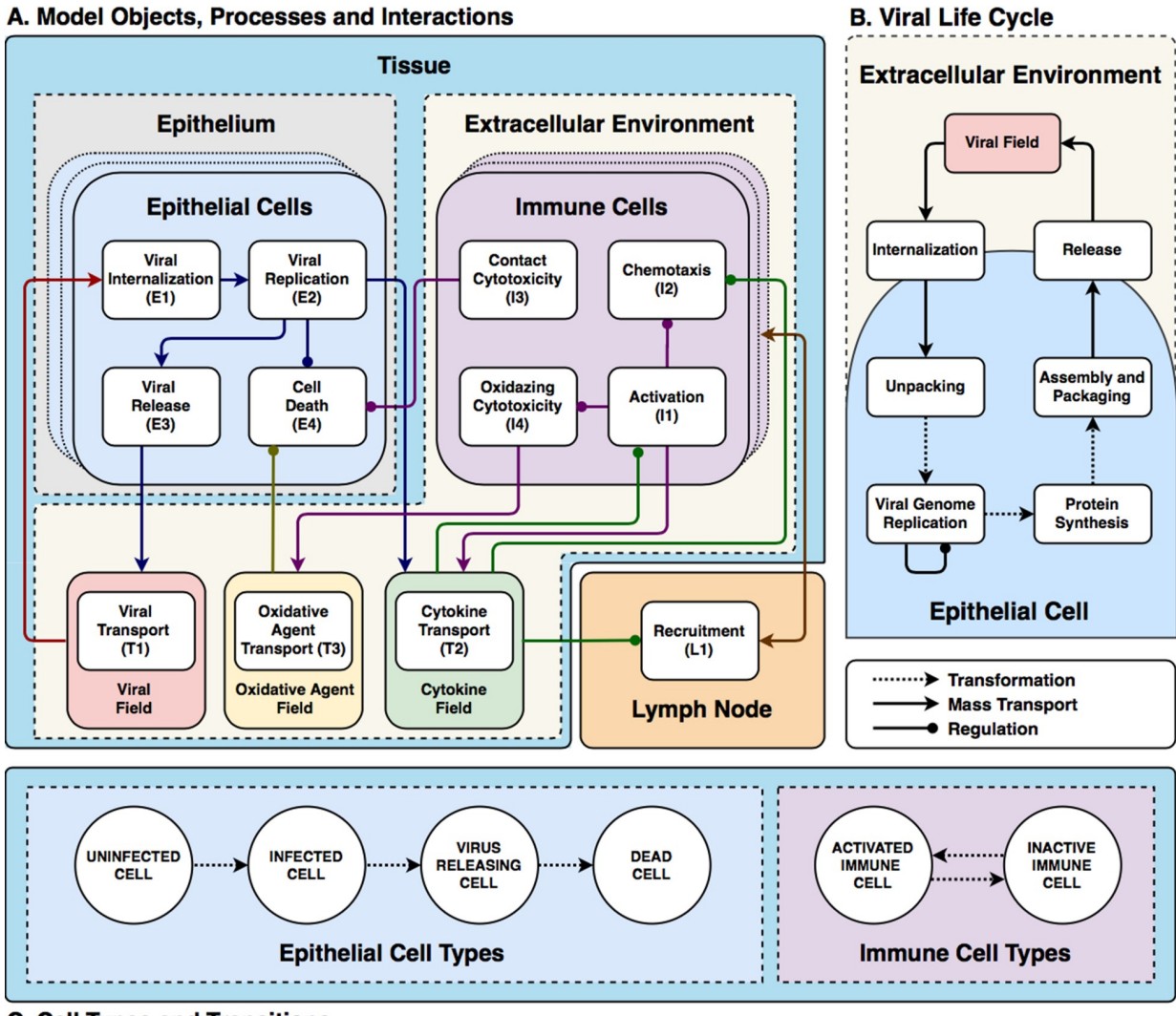

**Fig 2. Full model schematic.** (A) Model objects, processes and interactions: Conceptual model of an epithelial tissue and lymph node. Schematic representation of the model objects, processes and interactions. Epithelial and immune cells refer to the two main classes of cells. Interactions occur within an extracellular environment, and a compartmental model of a lymph node controls immune-cell recruitment to the tissue. Together the epithelial-cell, extracellular-environment and immune-cell components represent the epithelial tissue. Each model object has associated processes that dictate its states and behaviors. Epithelial-cell processes include viral internalization (**E1**), viral replication (**E2**), viral release (**E3**) and cell death (**E4**). Immune cell processes include activation (**I1**), chemotaxis (**I2**), contact cytotoxicity (**I3**) and oxidative cytotoxicity (**I4**). *Activated* immune cells participate in oxidative cytotoxicity (**I4**) and secrete oxidative agents into the oxidizing-agent field (**T3**). Activated cells become inactive after 1 hour. The virus field (**T1**), cytokine field (**T2**) and oxidizing-agent field (**T3**) describe spatially-varying densities of extracellular components. Field processes describe diffusive transport and clearance of material in the extracellular environment and activated transport to the lymph nodes. The lymph node is a single-compartment model whose pro- or anti-inflammatory state specifies the recruitment or removal rate (**L1**) of immune cells in the epithelial tissue. The transport of cytokines to the lymph node promotes its proinflammatory state. (B) Viral Life Cycle: Interactions in the viral internalisation, replication and release models. Schematic representation of inputs, outputs and interactions between stages of the viral replication model. Extracellular viral particles (represented as continuous fields) are internalized by the viral internalization model and initiate the viral replication model. The main stages of the viral replication model are: unpacking, viral genome replication, protein synthesis and viral assembly and packaging. The output of the viral replication model passes to the viral release model, which transfers newly assembled viral particles from the cells into the extracellular environment. (C) Cell types and transitions. Epithelial cells are of type *uninfected* if they have not yet internalized any virus (**E1**). They are of type *infected* if they have internalized virus, but are not releasing virus into the virus field (viral release **E3** is inactive). They are of type *virus releasing* if they are releasing virus into the extracellular virus field (*i.e.*, viral release **E3** is activated). Immune cells are initially *inactive* and do not participate in the oxidative cytotoxicity (**I4**) or chemotax towards higher concentrations of the cytokine field (**I2**). They become *activated* when they experience activation (**I1**). In all panels, dashed arrows with barbed heads represent transformations, solid arrows with barbed heads represent transport and solid arrows with lollipop heads represent regulation.

immune-cell mediated cell killing, rather than any particular immune cell phenotype. We omit macrophages, neutrophils and their phagocytosis and signaling, which can be represented using simple extensions of the immune-cell type. We do not model contact interactions between immune cells (*e.g.*, by T-cells and APCs). We also do not model tissue recovery, though we demonstrate examples of adding tissue recovery models in *Model extensions*.

We simulate extracellular-virus particle density as a continuum field and particle transport and clearance as continuous diffusion and decay. We approximate the discrete process of a cell's internalization of a virus particle by a stochastic virus internalization event (**E1**) determined by time elapsed, the local concentration of the virus field, and the number of available cell-surface receptors on the cell. We simplify the complexity of viral replication into four steps: unpacking, viral genome replication, protein synthesis and packaging/assembly (**E2**, Fig 2B) [7,52–54]. The subcellular kinetics of viral replication determine the rate of release of new virus particles into the extracellular environment (**E3**). To represent the combined effect of the many types of virus-induced cell death, each infected epithelial cell has a probability of dying that depends on the number of assembled viral particles inside the cell per unit time (**E4**).

We simplify the complex biochemistry of the many molecular signals active in epithelial tissues, which include chemokines, interferons and viral fragments, into a single generic extracellular cytokine field that acts both as a tissue-local and systemic pro-inflammatory signal. Infected epithelial cells and immune cells both secrete cytokine (**T2**). The cytokine field produces local immune effects such as activation (**I1**) and chemotaxis (**I2**) of immune cells. Activated immune cells can revert back to inactive immune cells when the cytokine signal ceases. The cytokine field also recruits immune cells to the tissue through long-distance signaling via the lymphatic system (**L1**). We model recruitment of immune cells to the simulation domain using an ordinary differential equation for the *immune signal* (*S*), which represents the balance between pro- and anti-inflammatory signaling and the delay due to antigen-presenting cell transport from the tissue through the lymphatic system to the lymph node and due to the time required for T-cell amplification. In the absence of infection, the lymph node maintains a small resident immune cell population in the tissue. Immune cells can cause epithelial cell death (**E4**) by three mechanisms: 1) contact cytotoxicity; 2) the bystander effect; and 3) through the release of an oxidative agent. Immune cells kill infected epithelial cells by contact cytotoxicity, in which case neighboring uninfected, infected and virus-releasing epithelial cells can also die through a bystander effect (**I3**). In regions of the tissue with high cytokine levels, immune cells secrete a diffusive oxidative agent to the environment (**T3**) that kills uninfected, infected and virus-releasing epithelial cells (**I4**).

## Results

We begin by presenting our base multicellular model of viral infection in an epithelial tissue, along with a simulation for a baseline set of parameters and basic analyses. We then explore the simulation's dependence on critical parameters that are important to the dynamics of acute primary viral infection in airway epithelial cells. All simulations and spatial, population and system-level metrics presented in this section follow the specifications in *Simulation Specifications*. We performed simulations using the open-source modeling environment Compu-Cell3D [55]. Instructions on how to run these simulations are provided in our publicly available repository at https://github.com/covid-tissue-models/covid-tissue-response-models/tree/master/CC3D/Models/BiocIU/SARSCoV2MultiscaleVTM.

We initialize the simulations with periodic boundary conditions parallel to the plane of the sheet and Neumann boundary conditions perpendicular to the plane of the sheet. Initially, the extracellular environment does not contain any extracellular virus, cytokines, oxidative agents

or immune cells. We introduce infection by creating a single infected epithelial cell at the center of the epithelial sheet, comparably to (but less than) initial conditions of similar works that model discrete cellular distributions [18,31]. To illustrate the full range of dynamics of viral infection in the presence of an immune response, we established a baseline set of parameters (Table 1) for which the immune response is strong enough to slow the spread of the infection, but insufficient to prevent widespread infection and death of all epithelial cells (Fig 3). While we have adjusted the parameters for the viral replication model to agree with reported time scales for SARS-CoV-2 replication *in vitro* [56], and we have selected parameter values in physiologically reasonable ranges, we have not attempted to match other model parameters to a specific tissue, virus or host species. Furthermore, this baseline parameter set is not unique with respect to its purpose, in that many sets of parameters can generate appreciable but insufficient inhibition of spread of infection (see *S12–S17 Figs*). Rather, as is shown in subsequent sections, this parameter set presents emergent dynamics of a theoretical virus and host immune response near, but not in, a regime of successful prevention of widespread infection, which is critical to showing the effects of underlying mechanisms on emergent dynamics and resulting outcomes.

The infected cell immediately starts releasing cytokines into the extracellular environment. After an incubation period (~150 minutes, 2 ½ hours), the first infected epithelial cell (green) transitions from infected to virus releasing (red), and starts releasing viruses into the extracellular environment. Initial release of extracellular virus causes additional epithelial cells to become infected. Release of cytokines leads to delayed addition of immune cells to the simulation domain (Fig 3D). By 4000 minutes (67 hours, 2 ¾ days), the number of infected cells increases 10-fold and the epithelial cells start undergoing virally-induced death as the infection spreads radially outward from the initial site. The increase in the number of infected cells and the local cytokine concentration is accompanied by a similar increase in the local immune cell population. By 8000 minutes (133 hours, 5 ½ days), the number of dead epithelial cells around the initial infection site increases sharply. Following this phase of rapid cell death, the number of infected, virus-releasing and dead epithelial cells continues to increase exponentially but at a slower rate. This transition results in the formation of an annular region of infected cells spreading radially outwards from the initial infection site (Fig 3A), analogous to the Fisher equation for deterministic front propagation [58]. Total extracellular virus and cytokine continue to increase exponentially. The increase in cytokine results in continued recruitment of additional immune cells. By 16000 minutes (267 hours, 11 days), the number of uninfected epithelial cells reaches zero and the number of infected and virus-releasing cells peaks. Despite the declining number of infected and virus-releasing epithelial cells, the delayed immune response continues to add immune cells to the tissue. After about 16000 minutes (267 hours, 11 days), the extracellular virus and the amount of cytokine decrease exponentially as the remaining virus-releasing epithelial cells die. By 20000 minutes (333 hours, 14 days), all epithelial cells die and many immune cells leave the tissue.

## Classification of spatiotemporal infection dynamics

The rate at which infection propagates and the strength (speed and amplitude) of the immune response depend on multiple model parameters. Interplay between these rates leads to a wide spectrum of qualitatively-distinct spatiotemporal dynamics. The virus-receptor binding affinity $k_{on}$ and the immune response delay coefficient $\beta_{delay}$ are critical parameters affecting the rate of infection of epithelial cells and the strength of the immune response, respectively. Larger $k_{on}$ corresponds to a higher rate of infection propagation (increasing $k_{on}$ increases the rate of internalization of extracellular viral particles into epithelial cells, see Eq (3) in *Models*

**Table 1. Parameter values in baseline parameter set.**

| Conversion Factors | Value | References / Justification |
|---|---|---|
| Simulation step $\Delta t$ | 1200.0 s | Selected for approximately 14 days of simulation time in 1k simulation steps |
| Lattice width | 4.0 μm | Selected according to cell diameter |
| Scale factor for concentration | $1\times10^{-14}$ mol | Selected for conversion from mol L$^{-1}$ to mol μm$^{-1}$ ($10^{-15}$) with 10x adjustment for fewer concentrations less than 1 |
| Simulation parameters | Value | References / Justification |
| Cell diameter | 12.0 μm | Selected according to typical epithelial cell size |
| Replication rate $r_{max}$ | $(1/12)\times10^{-3}$ s$^{-1}$ | Calibrated to timescale of SARS-CoV-2 [56] |
| Translating rate $r_t$ | $(1/18)\times10^{-3}$ s$^{-1}$ | Calibrated to timescale of SARS-CoV-2 [56] |
| Unpacking rate $r_u$ | $(1/6)\times10^{-3}$ s$^{-1}$ | Calibrated to timescale of SARS-CoV-2 [56] |
| Packaging rate $r_p$ | $(1/6)\times10^{-3}$ s$^{-1}$ | Calibrated to timescale of SARS-CoV-2 [56] |
| Release rate $r_s$ | $(1/6)\times10^{-3}$ s$^{-1}$ | Calibrated to timescale of SARS-CoV-2 [56] |
| Scale factor for number of mRNA per infected cell $mRNA_{avg}$ | 1000 cell$^{-1}$ | Selected for average production of 2000 virions per cell before death per influenza [69] |
| Viral dissociation coefficient $r_{half}$ | 2000 | Derived from $mRNA_{avg}$, $r_{max}$ and $r_t$ |
| Viral diffusion coefficient $D_{vir}$ | 0.01 μm$^2$ s$^{-1}$ | Selected according to sensitivity analysis and estimated from physiological ranges for lung mucus[1] |
| Viral diffusion length $\lambda_{vir}$ | 36 μm | Selected according to sensitivity analysis |
| Viral decay rate $\gamma_{vir}$ | $7.71\times10^{-6}$ s$^{-1}$ | Derived from $\lambda_{vir}$ and $D_{vir}$ |
| Cytokine diffusion coefficient $D_{cyt}$ | 0.16 μm$^2$ s$^{-1}$ | [70,71] (IL-2 cytokine) |
| Cytokine diffusion length $\lambda_{cyt}$ | 100 μm | [70] (IL-2 cytokine) |
| Cytokine decay rate $\gamma_{cyt}$ | $1.32\times10^{-5}$ s$^{-1}$ | Derived from $\lambda_{cyt}$ and $D_{cyt}$ |
| Maximum cytokine immune secretion rate $\sigma_{cyt}(immune\ activated)$ | $3.5\times10^{-4}$ pM s$^{-1}$ | Estimated as 1/10 of $\sigma_{cyt}(infected)$ |
| Immune secretion midpoint $V_{cyt}(immune\ activated)$ | 1 pM | [71] |
| Cytokine immune uptake rate $\omega_{cyt}(immune\ activated)$ | $3.5\times10^{-4}$ pM s$^{-1}$ | [70] |
| Maximum cytokine infected cell secretion rate $\sigma_{cyt}(infected)$ | $3.5\times10^{-3}$ pM s$^{-1}$ | [70] |
| Infected cell cytokine secretion mid-point $V_{cyt}(infected)$, $V_{cyt}(virus\ releasing)$ | 0.1 | Chosen from typical simulation values of assembled virus. Values stay around 0.1 and increase as infection progresses |
| Cytokine secretion Hill coefficient $h_{cyt}$ | 2 | Selected for simplest form with inflection of model response |
| Immune cell cytokine activation $EC50_{cyt,act}$ | 10 pM | [70] |
| Immune cell equilibrium bound cytokine $EQ_{ck}$ | 210 pM | Chosen to be 2.1 x $EC50_{cvt,act}$ |
| Immune cell bound cytokine memory $\rho_{cyt}$ | 0.99998 s$^{-1}$ | Derived from $\omega_{cyt}(immune\ activated)$ and $EQ_{ck}$ |
| Immune cell activated time | 10 h | [72] |
| Oxidation Agent diffusion coefficient $D_{oxi}$ | 0.64 μm$^2$ s$^{-1}$ | Selected to be 4 x $D_{cyt}$ to model high diffusivity relative to IL-2 |
| Oxidation Agent diffusion length $\lambda_{oxi}$ | 36 μm | Selected to be 3 cell diameters to model high reactivity |
| Oxidation Agent decay rate $\gamma_{oxi}$ | $1.32\times10^{-5}$ s$^{-1}$ | Derived from $\lambda_{oxi}$ and $D_{oxi}$ |
| Immune cell oxidation agent secretion rate $\sigma_{oxi}$ | $3.5\times10^{-3}$ pM s$^{-1}$ | Selected to be the same as $\sigma_{cyt}(infected)$ |
| Immune cell $C_{cyt}$ threshold for Oxidation Agent release $\tau_{oxi}^{sec}$ | 10 A.U. = 1.5625 pM | Selected according to sensitivity analysis |

*(Continued)*

**Table 1.** (Continued)

| Conversion Factors | Value | References / Justification |
|---|---|---|
| Tissue cell $C_{oxi}$ threshold for death $\tau_{oxi}^{death}$ | 1.5 A.U. = 0.234375 pM | Selected according to sensitivity analysis |
| Initial density of unbound cell surface receptors $R_o$ | 200 cell$^{-1}$ | Selected for potential limiting factor (availability of receptors) from typical simulation extracellular virus field values |
| Virus-receptor association affinity $k_{on}$ | $1.4 \times 10^4$ M$^{-1}$s$^{-1}$ | [73,74] |
| Virus-receptor dissociation affinity $k_{off}$ | $1.4 \times 10^{-4}$ s$^{-1}$ | [73,74] |
| Infection threshold | 1 | Calibrated to timescale of SARS-CoV-2 [56] |
| Uptake Hill coefficient $h_{upt}$ | 2 | Selected for simplest form with inflection of model response |
| Uptake characteristic time constant $\alpha_{upt}$ | 20 min | Selected to be the same as $\Delta t$ |
| Virally-induced apoptosis Hill coefficient $h_{apo}$ | 2 | Selected for simplest form with inflection of model response |
| Virally-induced apoptosis dissociation coefficient $V_{apo}$ | 100 | Selected according to sensitivity analysis |
| Virally-induced apoptosis characteristic time constant $\alpha_{apo}$ | 20 min | Selected to be the same as $\Delta t$ |
| Immune cell activation Hill coefficient $h_{act}$ | 2 | Selected for simplest form with inflection of model response |
| Immune response add immune cell coefficient $\beta_{add}$ | 1/1200 s$^{-1}$ | Selected for sensitivity analysis of $\beta_{delay}$ |
| Immune response subtract immune cell coefficient $\beta_{sub}$ | 1/6000 cell$^{-1}$ s$^{-1}$ | Selected according to $\beta_{add}$ for 5 resident immune cells (mean of all immune cell counts per epithelial cell from Control in [22] applied to 900 epithelial cells) |
| Immune response delay coefficient $\beta_{delay}$ | $1.2 \times 10^6$ s | Selected according to sensitivity analysis |
| Immune response decay coefficient $\beta_{delay}$ | 1/12000 s$^{-1}$ | Selected for sensitivity analysis of $\beta_{delay}$ |
| Immune response cytokine transmission coefficient $\alpha_{sig}$ | 0.5 | Selected for sensitivity analysis of $\beta_{delay}$ |
| Immune response probability scaling coefficient $\alpha_{immune}$ | 0.01 | Selected for sensitivity analysis of $\beta_{delay}$ |
| Number of immune cell seeding samples $n_{seeding}$ | 10 | Selected for sensitivity analysis of $\beta_{delay}$ |
| Initial target volume | 64 $\mu$m$^3$ | Derived from cell diameter and lattice width |
| Lambda volume $\lambda_{volume}$ | 9 | Selected for acceptable deformation of immune cells |
| Initial number of immune cells | 0 | Selected to demonstrate model feature of resident immune cells |
| Lambda chemotaxis $\lambda_{chemotaxis}$ | 1 | Selected for appreciable chemotaxis without excessive cell deformation |
| Intrinsic Random Motility $\mathcal{H}^*$ | 10 | [68] |
| Contact coefficients $J$ (all interfaces) | 10 | Selected comparably to [68] for low adhesion immune cell-immune cell and immune cell-medium interfaces |

[1] The diffusivity in water for a virus of radius 0.1 microns like SARS-CoV-2 according to Stokes-Einstein is about 3 microns$^2$/s. The average steady-shear viscosity for lung mucus varies significantly and is shear thinning, but in the more viscous regions is found to vary for frequencies between $10^{-4}$ and 102 Hz, spanning viscosity values as high as 103 Pa-s and as low as $10^{-2}$ Pa-s. In general, at low shear rates, the viscosity of human mucus is as high as 104–106 times that of water [75]. Thus the minimal diffusion constant possible would be 3 x $10^{-6}$ microns$^2$/s and the maximal rate in water would be 3 microns$^2$/s. 0.01 microns$^2$/s is a reasonable geometric interpolation.

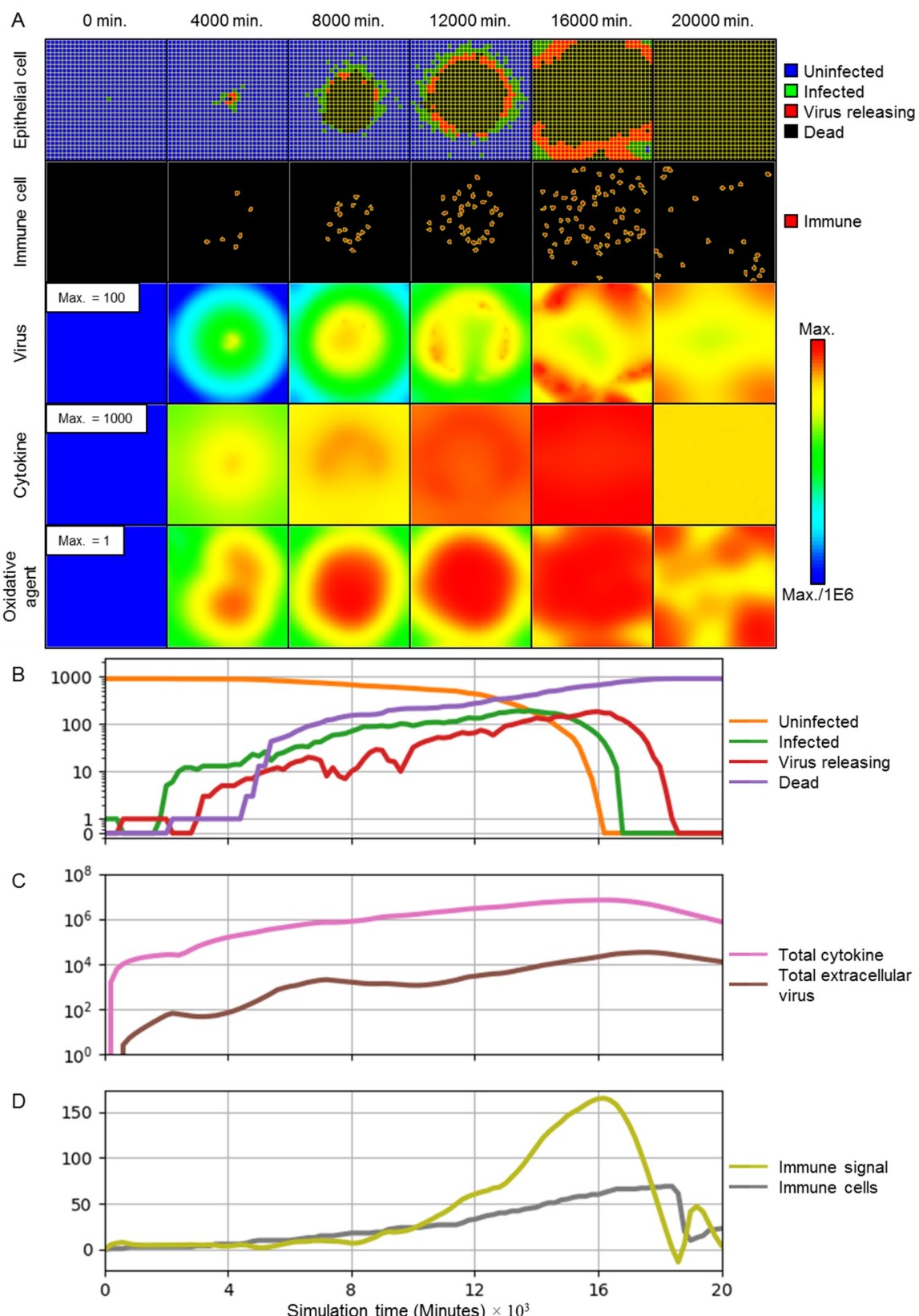

**Fig 3. Simulation of the progression of infection in a patch of epithelial tissue of size 360 μm x 360 μm starting from a single infected cell for a representative simulation using the baseline parameters given in Table 1.** (A) Snapshots of spatial configuration *vs* time, showing progression of a simulated infection. Columns, left to right: 0 minutes (time of initial infection), 4000 minutes (67 hours, 2 ¾ days) after infection, 8000 minutes (133 hours, 5 ½ days), 12000 minutes (200 hours, 8 ⅓ days), 16000 minutes (267 hours, 11 days), and 20000 (333 hours, 14 days) minutes. First row: epithelial-cell layer composed of uninfected (blue), infected (green), virus-releasing (red) and dead epithelial cells (black). Second row: position of immune cells in the extracellular environment layer. Third row: concentration of extracellular virus field. Fourth row: concentration of extracellular cytokine field. Fifth row: concentration of extracellular oxidative agent field. Fields are shaded on a logarithmic scale: red corresponds to the chosen maximum value specified in the first panel and blue corresponds to six orders of magnitude lower than the maximum value; colors saturate for values outside this range. (B-D) Simulation time series. (B) Number of uninfected (orange), infected (green), virus-releasing (red) and dead (purple) epithelial cells *vs* time on a logarithmic scale (0 values are overlaid at a non-logarithmic tick for clarity). (C) Total extracellular cytokine (magenta) and total extracellular virus (brown) *vs* time on a logarithmic scale. (D) Value of the immune recruitment signal $S$ (yellow) and number of immune cells (grey) *vs* time on a linear scale. Simulations use periodic boundary conditions in the plane of the epithelial sheet, and Neumann conditions [57] normal to the epithelial sheet.

*and methods*). Larger $\beta_{delay}$ corresponds to weaker immune response (decreasing $\beta_{delay}$ increases the strength of immune-cell recruitment, see Eqs (12)–(14) in *Models and methods*).

Varying these two parameters around the baseline simulation values yields six patterns of spatiotemporal infection dynamics, ranging from unopposed infection to clearance (Fig 4). We defined these classes based on the transient dynamics and the final state of the simulation at 20000 minutes (333 hours, 14 days). We terminated the simulations at 20000 minutes (333 hours, 14 days) because we assume that, in real tissues, additional adaptive immune responses at this time generally lead to rapid viral clearance. As a result, any epithelial cells uninfected at the end of the simulation would likely remain uninfected if we included these additional immune mechanisms. We define the six patterns (*classes*) of infection dynamics as follows:

**No immune response:** a limiting case (corresponding to *in vitro* and organoid culture experiments on viral infection, which lack immune cells) that serves as a reference simulation showing the spread of unopposed infection. When the cellular immune response is absent, an infection front travels across the epithelium until all epithelial cells have died due to intracellular virus (Fig 4A and S1 Fig).

**Widespread infection:** when the immune response is weak (large $\beta_{delay}$) or the rate of infection propagation is large (large $k_{on}$), the immune cells kill enough infected epithelial cells to reduce both the concentration of extracellular virus and the propagation of the infection front. However, at the end of the simulation no uninfected epithelial cells survive (Figs 3 and 4B).

**Slowed infection:** for moderate immune response (moderate $\beta_{delay}$) and a moderate rate of infection propagation (moderate $k_{on}$), both uninfected and infected epithelial cells and some extracellular virus remain at the end of the simulation (Fig 4C). These cases are functionally distinct from widespread infection, since even a single remaining uninfected epithelial cell could initiate tissue regeneration. In most cases of slowed infection, the numbers of infected cells and the extracellular virus continue to increase. A special case of slowed infection occurs when oscillations in the amount of virus (S2A Fig).

**Containment:** for strong immune response (small $\beta_{delay}$) and low to moderate rate of infection propagation (moderate $k_{on}$), a few infected and virus-releasing cells are present for most of the simulation. However, the immune cells eventually kill all infected and virus-releasing epithelial cells. At the end of the simulation, no infected or virus releasing cells remain, while uninfected cells survive and some extracellular virus remains in the extracellular environment (Fig 4D).

**Recurrence:** for strong immune response (small $\beta_{delay}$) and a fast infection propagation (large $k_{on}$), relatively few epithelial cells become infected early in the simulation. All infected and virus-releasing epithelial cells die. However, the remaining extracellular virus infects additional epithelial cells later on, restarting the spread of infection (Fig 4E).

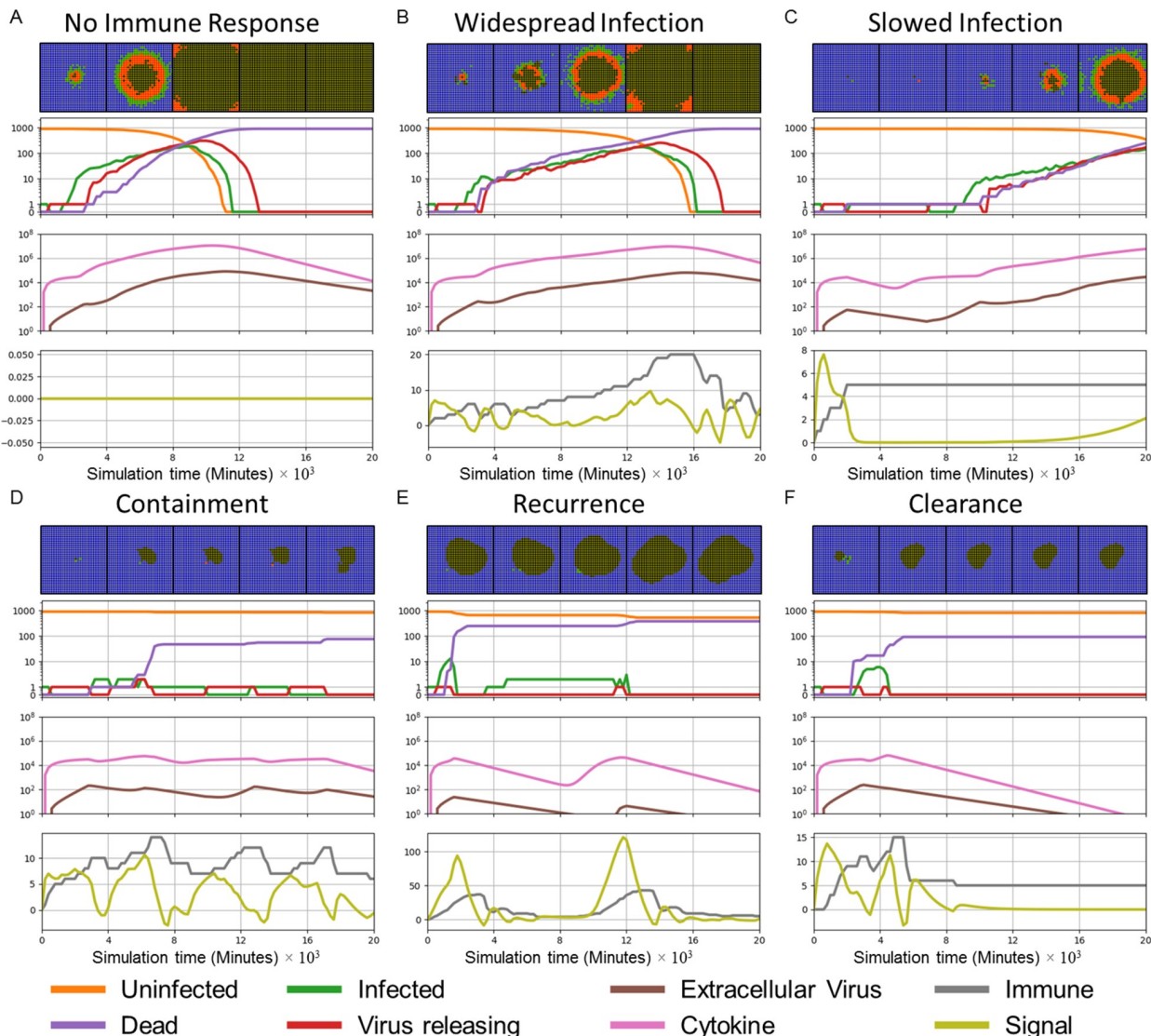

**Fig 4. Patterns (classes) of spatiotemporal infection dynamics.** First row: snapshots of spatial configurations of the epithelial cells. Color coded: uninfected (blue), infected (green), virus releasing (red), dead (black). TImes from left to right 4000 minutes (67 hours, 2 ¾ days), 8000 minutes (133 hours, 5 ½ days), 12000 minutes (200 hours, 8 ⅓ days), 16000 (267 hours, 11 days) and 20000 minutes (333 hours, 14 days). The right border of each snapshot aligns with the corresponding time in the time series. Second row: number of uninfected (orange), infected (green), virus-releasing (red) and dead (purple) epithelial cells *vs* time on a logarithmic scale (with 0 included for clarity). Third row: total extracellular cytokine (magenta) and total extracellular virus (brown) *vs* time on a logarithmic scale. Fourth row: value of the immune recruitment signal $S$ (yellow) and number of immune cells (grey) *vs* time on a linear scale. (A) No immune response: infection propagates unopposed until all epithelial cells have died from intracellular virus. (B) Widespread infection: weak immune response slows propagation of the infection, but no uninfected cells survive at the end of the simulation. (C) Slowed infection: uninfected and infected epithelial cells coexist at the end of the simulation. (D) Containment of infection: no infected or virus-releasing epithelial cells remain, uninfected cells survive and virus remains in the extracellular environment at the end of the simulation. (E) Recurrence: the number of infected and virus releasing epithelial cells goes to zero, but persistent extracellular virus infects new epithelial cells later on. (F) Clearance: the number of infected and virus-releasing epithelial cells goes to zero and the level of extracellular virus is negligible at the end of the simulation. The model in (A) omits the immune response (components **L1, I1-I4**). All parameter values are as in Table 1 and Fig 3 except for $k_{on}$ and $\beta_{delay}$ (S1 Table).

**Clearance:** for strong immune response (small $\beta_{delay}$) and a slow infection propagation (small $k_{on}$), the number of infected and virus-releasing epithelial cells goes to zero without recurrence and the extracellular virus drops below a threshold (of 1/900 per cell in our analysis), rendering the frequency of recurrence negligible (Fig 4F). A special case of clearance

(**Failure to infect**) occurs when the initially infected epithelial cells fail to infect any other epithelial cells (S2B Fig).

To quantitatively characterize simulation results, we measured the number of uninfected, infected, virus-releasing and dead epithelial cells, the total number of immune cells, the number of activated immune cells, the total amount of extracellular virus (integral over the virus field), the total diffusive cytokine (integral over cytokine field), the maximum total extracellular virus (over all simulation time) and the maximum total diffusive cytokine (over all simulation time). Fig 4 shows these quantitative metrics, together with a series of spatial configurations for all model components, corresponding to each pattern of infection dynamics.

In the absence of an immune response, spread of the infection follows the diffusion of extracellular virus, resulting in concentric rings of different types of epithelial cells (see Fig 4A top row for epithelial cell types, S1B Fig shows the extracellular virus field). The viral propagation front, where uninfected epithelial cells transition to infected epithelial cells, moves radially outwards from the initial site of infection. Due to the stochastic internalization events, the front's outer contour is diffuse, with scattered infected epithelial cells ahead of the front, followed by a dense rim of infected and virus-releasing epithelial cells and a core of dead epithelial cells at the center. Similar waves, with a slower speed, are seen in the other cases where infection has occurred (Fig 4B and 4C top row).

## Stronger immune response and lower rates of virus internalization promote containment of infection

To explore the effects of the rate of virus internalization and the strength of the immune response, we performed a multidimensional parameter sweep of the virus-receptor association affinity $k_{on}$ and immune response delay coefficient $\beta_{delay}$. Variations in virus receptor association affinity represent factors that affect the binding affinity of cellular viral receptors (*e.g.*, ACE2 and TMPRSS-2 in the case of SARS-CoV-2) with a virus (*e.g.*, mutations in viral coat protein or drugs to block viral entry). Variations in immune response delay coefficient represent factors that affect the strength of the systemic immune response (*e.g.*, anti-inflammatory corticosteroids, IL-7 treatment or age, since older individuals tend to have slower adaptive immune responses)[36].

We ran ten simulation replicas for each parameter set, increasing and decreasing the baseline parameter values 10-fold and 100-fold (Figs 5–7). For each simulation replica, we examined the number of uninfected epithelial cells (Fig 5), the number of infected epithelial cells (Fig 6), the total extracellular virus (Fig 7), the number of dead epithelial cells (S3 Fig), the number of virus-releasing cells (S4 Fig) and the number of immune cells (S5 Fig). We identified regions of the parameter space where all ten simulation replicas resulted in either containment/clearance (green-shaded subplots) or widespread infection (orange-shaded subplots). In the intermediate regions (unshaded subplots) different replicas for the same set of parameters exhibited other (and sometimes multiple) classes of dynamics.

For large $k_{on}$ and large $\beta_{delay}$ (Figs 5–7, orange-shaded regions), all simulation replicas result in widespread infection and variability between simulation replicas is small. In this region, the initial release of virus into the extracellular environment results in a rapid increase in the number of infected and virus releasing epithelial cells early during the simulation, between 0 and 2000 minutes (0 to 33 hours, 0 to 1 ½ days) (Fig 6 and S4 Fig). Between 5000 and 20000 minutes (83 to 333 hours, 3 ½ to 14 days), the number of uninfected epithelial cells rapidly decays to zero. As in all simulation replicas with a large $\beta_{delay}$, the immune recruitment signal (S6 Fig) is less responsive to the cytokine signal produced by infected and virus-releasing epithelial cells and no significant recruitment of immune cells occurs throughout the simulations (S5 Fig). The

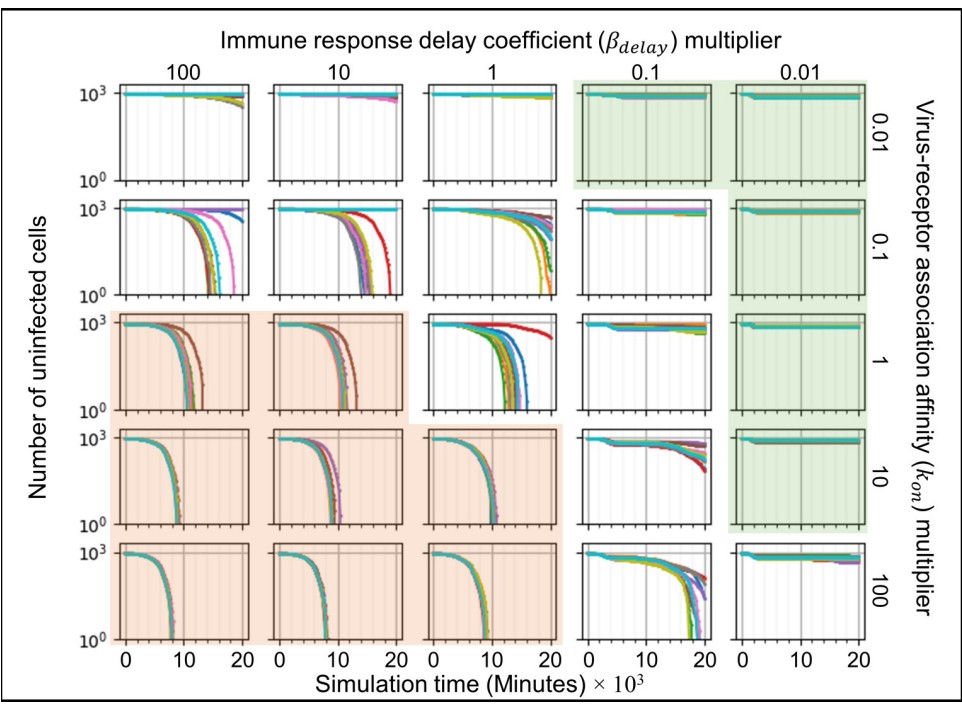

**Fig 5. Sensitivity analysis of the number of uninfected epithelial cells vs time for variations in virus-receptor association affinity $k_{on}$ and immune response delay coefficient $\beta_{delay}$, showing regions with distinct infection dynamics.** Logarithmic pairwise parameter sweep of the virus-receptor association affinity $k_{on}$ and the immune response delay $\beta_{delay}$ (×0.01,× 0.1,× 1,× 10,× 100) around their baseline values, with ten simulation replicas per parameter set (all other parameters have their baseline values as given in Table 1). The number of uninfected epithelial cells for each simulation replica for each parameter set, plotted on a logarithmic scale, *vs* time displayed in minutes. See Fig 4 for the definitions of the classes of infection dynamics.

number of virus-releasing epithelial cells peaks around 7500 minutes (125 hours, 5 days), the level of extracellular virus peaks around 9000 minutes (150 hours, 6 ¼ days) and the number of dead epithelial cells peaks around 10000 minutes (167 hours, 7 days, S3 Fig). With no remaining uninfected epithelial cells to infect, all remaining infected epithelial cells become virus-releasing epithelial cells, which then die, causing the total amount of extracellular virus to decrease.

For small $k_{on}$ and small $\beta_{delay}$ (Figs 5–7, green-shaded subplots), all simulation replicas result in either clearance or containment and variability between simulation replicas is small. Initial release of virus to the extracellular environment results in a small change in the number of uninfected and infected epithelial cells. As in all simulation replicas with a small $\beta_{delay}$, the immune recruitment signal (S6 Fig) is very sensitive to the cytokine produced by virus-releasing epithelial cells, resulting in rapid recruitment of immune cells and an early first peak in the population of immune cells around 2000 minutes (33 hours, 1 ⅓ days, S5 Fig). In these simulations, the increase in the number of immune cells is followed by a rapid increase in the number of dead epithelial cells (S3 Fig). The number of immune cells and the number of dead epithelial cells peak around the same time (2000 minutes, 33 hours, 1 ⅓ days), after which the number of immune cells decreases. With no remaining virus-releasing epithelial cells, the total extracellular virus decays until the infection is cleared. Increasing $\beta_{delay}$ primarily increases the time when all infected epithelial cells have died.

For moderate to high $k_{on}$ and moderate to low $\beta_{delay}$ (right unshaded subplots in Figs 5–7), the rate of new infection nearly balances the rate of elimination of infected and virus-releasing

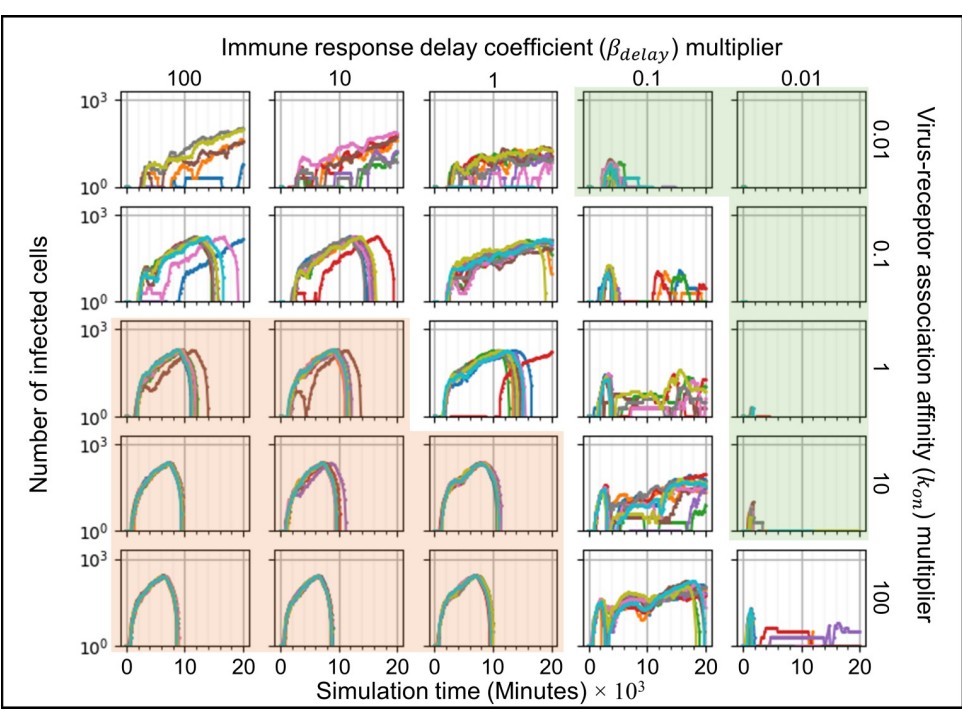

**Fig 6. Sensitivity analysis of the number of infected epithelial cells vs time for variations in virus-receptor association affinity $k_{on}$ and immune response delay coefficient $\beta_{delay}$, showing regions with distinct infection dynamics.** Same parameter sweep as Fig 5. The number of infected epithelial cells for each simulation replica for each parameter set, plotted on a logarithmic scale, *vs* time displayed in minutes. See Fig 4 for the definitions of the classes of infection dynamics.

epithelial cells, resulting in replicas showing clearance, contaminant, recurrence and slowed infection for the same parameter values, with very few cases of widespread infection. The initial release of virus into the extracellular environment by the first virus-releasing epithelial cell infects a moderate number of uninfected epithelial cells. The resulting cytokine secretion elicits a moderate to high response of the immune recruitment signal (S6 Fig) and number of immune cells (S5 Fig). The early recruitment of immune cells leads to many epithelial cells dying before 4000 minutes (66 hours, 2 ⅔ days) (S3 Fig). For a high probability of virus internalization (high $k_{on}$), even low amounts of extracellular virus are sufficient to cause recurrence. For moderate to low $k_{on}$ and moderate to high $\beta_{delay}$ (upper left unshaded subplots in Figs 5–7), rate of new infection of epithelial cells is slightly faster than the immune system's response, resulting in a combination of widespread infection, slowed infection and containment, and a few cases of clearance. The immune system is only moderately responsive to the cytokine signal, resulting in a slow to moderate increase in the immune recruitment signal (S6 Fig) and in the number of immune cells (S5 Fig). Cases of clearance and containment occur for a smaller final number of dead epithelial cells (S3 Fig) compared to previously discussed cases.

### Even moderate inhibition of genomic replication by antiviral therapies significantly reduces the spread of infection, but only when initiated soon after infection

Optimal therapeutic use of antiviral drugs requires considering the relationship between molecular *efficacy* (how effectively the drug blocks a particular aspect of the viral life cycle at saturation concentration), *potency of therapy* (the effect of the drug at a molecular level at a

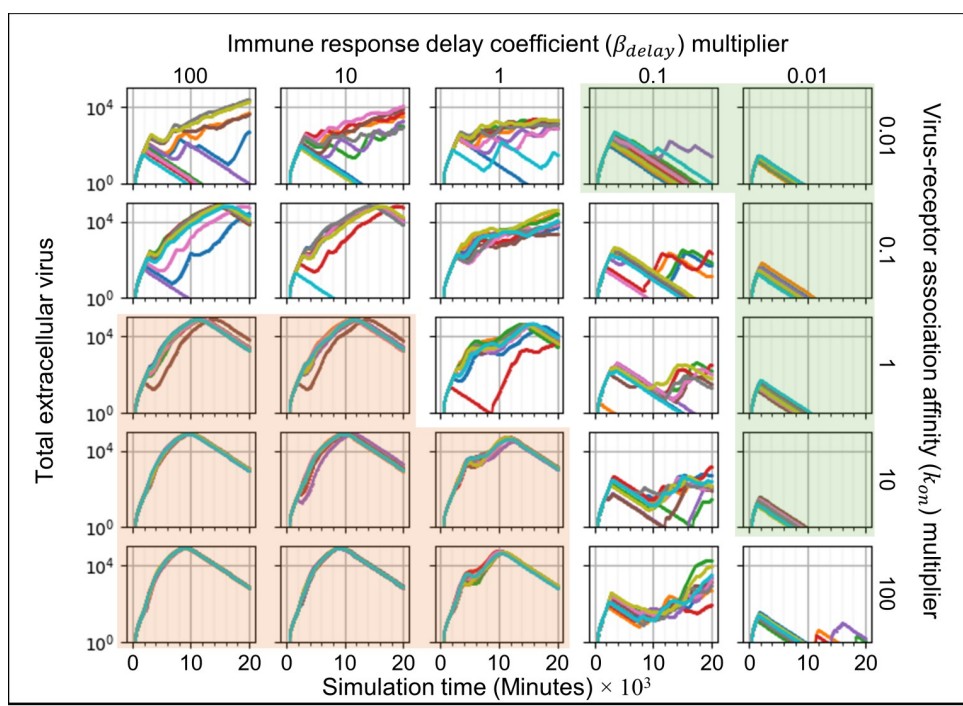

**Fig 7. Sensitivity analysis of the total amount of extracellular virus vs time for variations in virus-receptor association affinity $k_{on}$ and immune response delay coefficient $\beta_{delay}$, showing regions with distinct infection dynamics. Same parameter sweep as Fig 5.** The total amount of extracellular virus for each simulation replica for each parameter set, plotted on a logarithmic scale, *vs* time displayed in minutes. See Fig 4 for the definitions of the classes of infection dynamics.

given dose) and clinical *effectiveness* (how well the drug reduces the severity or duration of the infection), as well as the tradeoff between side effects and bioavailability. One drug might have moderate efficacy but have few side effects. Another drug might have high efficacy, but have severe side effects at high doses that limit the maximum tolerated dose or use of even moderate doses in prophylaxis. A drug might also have a combination of beneficial and adverse effects (*e.g.*, it might reduce viral replication early in infection, but also be immunosuppressive) [13,37]. Antiviral drugs like Tamiflu retain their ability to block aspects of the viral life cycle (efficacy), but become much less clinically effective as the time before treatment increases: (in adults Tamiflu is most effective when given within 48 hours after exposure and thus is often used prophylactically) [38].

In this section we use our model to show the trade-offs between time-of-use and potency of a drug that targets viral genome replication in a host cell. Several antiviral medications for RNA viruses reduce the net viral replication rate by inhibiting synthesis of viral RNA by the viral RNA polymerase. We focus on RNA-synthesis blockers in this paper because viral genome synthesis exponentially increases the production rate of viruses per cell, while the other stages of viral replication have linear amplification effects (see Eqs (6)–(9) in *Models and methods*).

To simulate the effects of treatment that targets RNA synthesis using different drug efficacies and times of administration, we generated a series of simulations in which we reduced $r_{max}$, the replication rate of genomic material in the viral replication model (Eq (7) in *Models and methods*), by different amounts and at different times in the simulation. The "viral replication multiplier" represents the potency of the treatment, the factor by which $r_{max}$ is reduced

(either a low dose with high efficacy, or a high dose with a less efficacy). The "time delay of application" is the simulation time at which $r_{max}$ is reduced, which corresponds to the time after infection at which the treatment is administered. To characterize therapeutic effectiveness, we distinguished three classes of simulation outcomes:

**Positive outcomes**: effective treatment, where at least 50% of the epithelial cells remain uninfected at the end of the simulation (green-shaded subplots).

**Negative outcomes**: ineffective treatment, where less than 10% of the epithelial cells remain uninfected at the end of the simulation (orange-shaded subplots).

**Intermediate outcomes**: partially effective treatment, where between 10–50% of the epithelial cells remain uninfected at the end of the simulation (unshaded or intermediate-shaded subplots).

To characterize how the potency and time of initiation of treatment affect the dynamics of the simulation and treatment effectiveness, we examined the time courses of the number of uninfected epithelial cells (Fig 8), virus-releasing epithelial cells (Fig 9), the total amount of extracellular virus (Fig 10), the number of dead epithelial cells (S7 Fig), the number of infected cells (S8 Fig), the number of immune cells (S9 Fig) and the immune response state variable (S10 Fig). Intensity of green indicates the percent of simulation replicas that produced positive outcomes for a given set of parameters. Intensity of orange indicates the percent of simulation replicas that produced negative outcomes.

When the treatment is given early, while the level of extracellular virus is increasing rapidly and exponentially (before 6000 minutes, 100 hours, 4 days) after infection, most of the simulation replica outcomes are positive, showing effective treatment (Figs 9–11, green-shaded subplots). If the drug is administered prophylactically or very soon after infection (at 0 minutes) the treatment potency needs to be only 25% to achieve mostly positive outcomes (effective treatment). Increasing the time to treatment increases the potency required to achieve similar numbers of positive outcomes: the treatment is effective for a potency of at least 37.5% if administered by 4000 minutes (67 hours, 2 ¾ days), and at least 87.5% if administered by 6000 minutes (100 hours, 4 days). For all potencies greater than 12.5%, early intervention prevents significant increase in the number of virus-releasing cells (Fig 9, green-shaded subplots), and a small number of immune cells suffices to stop the spread of infection (S9 Fig, green-shaded subplots). In this region, delaying treatment results both in a higher level of extracellular virus (Fig 10, green-shaded subplots) and more dead epithelial cells at the end of simulation (Fig 8, green-shaded subplots). With inhibited viral replication in the infected epithelial cells, the extracellular virus decays until it is mostly cleared by the end of simulation (Fig 10). Variability between simulation replicas for a given parameter set increases with both decreasing potency and increasing time of initiation of treatment.

If the potency of the treatment is 12.5% (or less), most of the simulation replicas have negative outcomes (low effectiveness), even if the drug is administered prophylactically or soon after infection (at 0 minutes) (Figs 9–11, bottom row). In these cases, the time after infection at which the drug is given makes no significant difference to the treatment effectiveness. When the treatment is given late (time delay of application of at least 10000 minutes, 167 hours, 7 days), regardless of the potency of the drug, most simulation replicas have negative outcomes (Figs 9–11, orange-shaded regions). By the time of treatment, a significant number of epithelial cells have been infected (more than 10% in most cases–Fig 9, orange-shaded regions) and a significant amount of virus has been released into the extracellular environment (Fig 10, orange-shaded regions). In addition, a significant number of epithelial cells have died (more than 10% in most cases–S7 Fig, orange-shaded regions) and significant recruitment of immune cells has occurred (S9 Fig, orange-shaded regions). For higher treatment potency, the level of virus in the extracellular environment starts decreasing immediately after treatment,

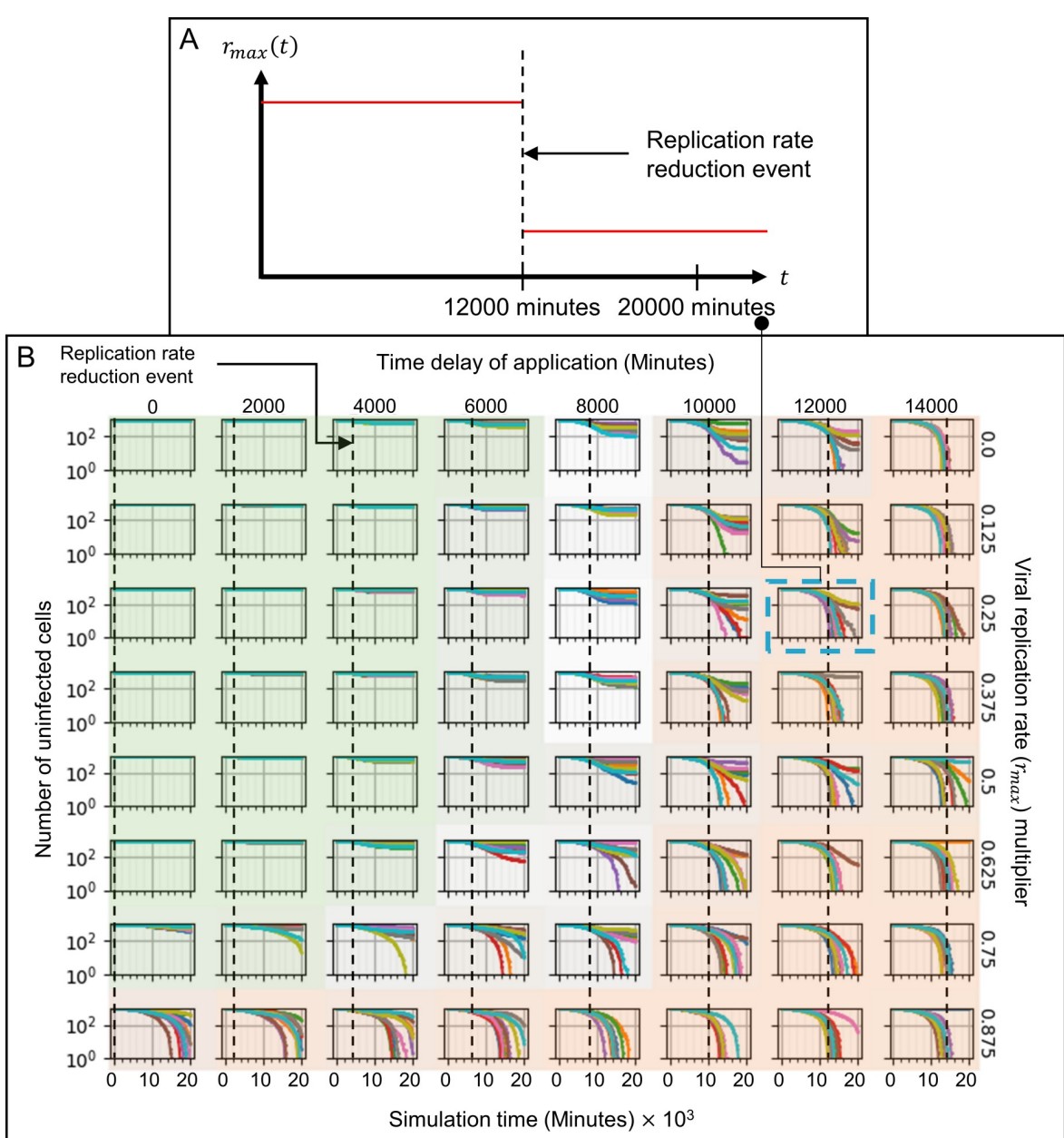

**Fig 8. Number of uninfected cells vs time in simulations of a hypothetical drug treatment reducing the viral genome (e.g. RNA for SARS-CoV-2) replication rate ($r_{max}$) as a function of treatment potency and time of initiation of treatment.** Drug therapy is administered at a fixed time after infection and remains activated for the duration of the simulation. (A) Sample treatment, showing the time course of $r_{max}$. $r_{max}$ is reduced by a multiplier which is one minus the potency of the drug at the given dose, 75% in (A), at a particular time of initiation of treatment (time delay of application), 12000 minutes (200 hours, 8 ⅓ days) in (A). (B) A parameter sweep of the potency of treatment (reduction in baseline viral replication rate $r_{max}$, vertical) and the time of treatment (dashed lines, horizontal) shows parameter regions where the majority of simulation replicas produce positive outcomes (green-shaded subplots), negative outcomes (orange-shaded subplots) and intermediate cases (intermediate shading or unshaded). Intensity of green and orange indicates the number of positive and negative outcome replicas for each parameter combination (treatment effectiveness). Green regions show that early intervention leads to positive outcomes (is effective) for most ranges of treatment potency, with high numbers of uninfected epithelial cells at the end of the simulation for almost all simulation replicas. Orange regions show that late interventions result in mostly negative outcomes (ineffective treatment) regardless of the potency, and that outcomes are more variable between replicas, with both positive and negative outcomes for most parameter sets. The number of uninfected epithelial cells for each simulation replica for each parameter set, plotted on a logarithmic scale, *vs* time displayed in minutes.

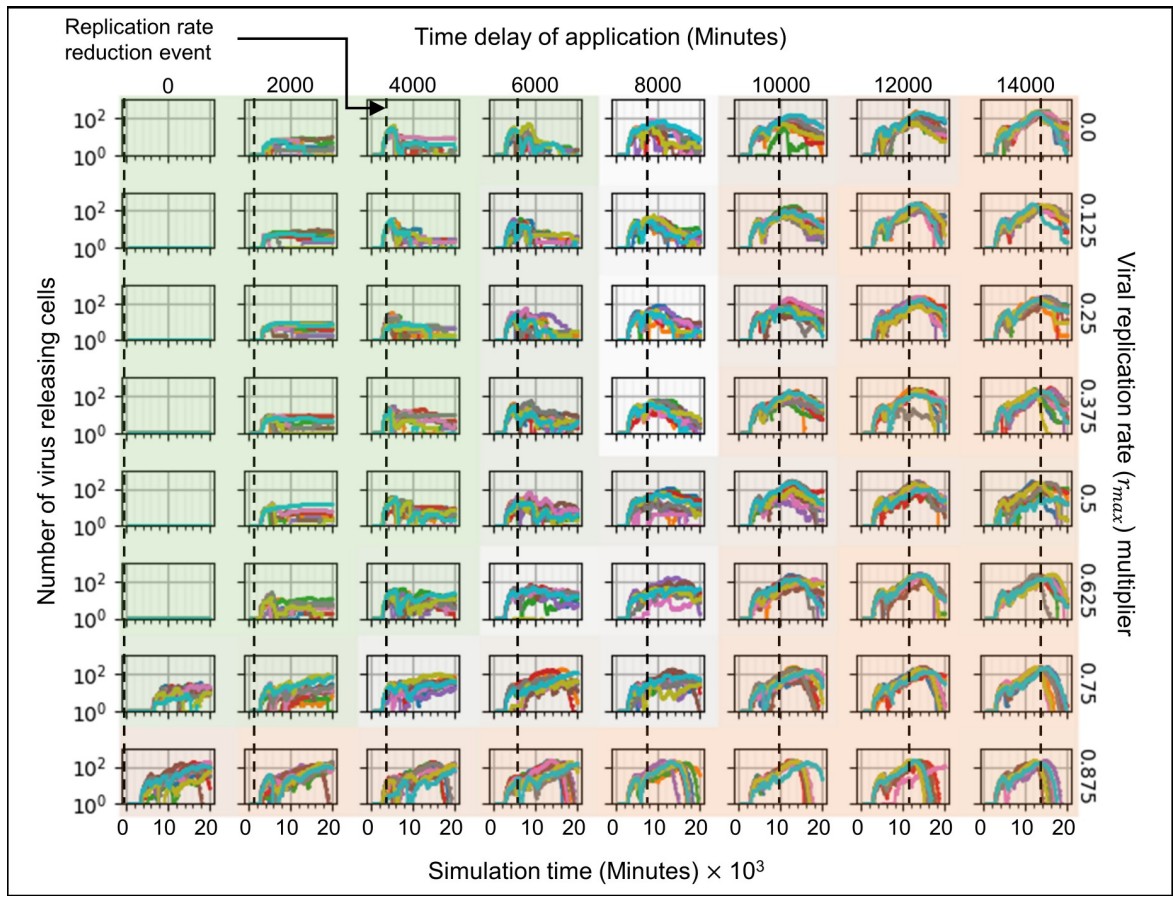

**Fig 9. Number of virus releasing cells vs time in simulations of a hypothetical drug treatment reducing the viral genome (e.g. RNA for SARS-CoV-2) replication rate ($r_{max}$) as a function of treatment potency (one minus the viral replication rate multiplier) and time of initiation of treatment.** The number of virus-releasing epithelial cells stays low when the intervention occurs early during infection (when the amount of extracellular virus is increasing rapidly), but continues to increase when the intervention occurs later (when the level of extracellular virus is at or near its maximum in the untreated case). Parameter values, axis types and time-scale and shading as in Fig 8.

even when a significant number of virus-releasing epithelial cells remain, indicating that viral replication inside cells has been significantly reduced. Later intervention also increases variability between simulation replicas and, although most simulation replicas have negative outcomes, the same set of parameter values produced two distinct qualitative outcomes (some more and some less favorable) for higher potency (S11 Fig, orange-shaded regions). Thus in a few cases, even late treatment can be effective.

When the treatment is given at intermediate times (times between 6000 and 10000 minutes, 100 to 167 hours, 4 to 7 days), most simulation replicas have intermediate outcomes. For potencies above 50%, the fraction of uninfected epithelial cells at the end of simulation is relatively high (around 50%) and the treatment is usually moderately effective (Fig 8). For potencies below 50%, the number of virus-releasing epithelial cells remains approximately constant or continues to increase after treatment (Fig 9) and significant levels of extracellular virus remain at the end of the simulation, and so in most cases the treatment is ineffective (Fig 10). In this regime, variability between outcomes for the same parameter values is higher than for potencies above 50%.

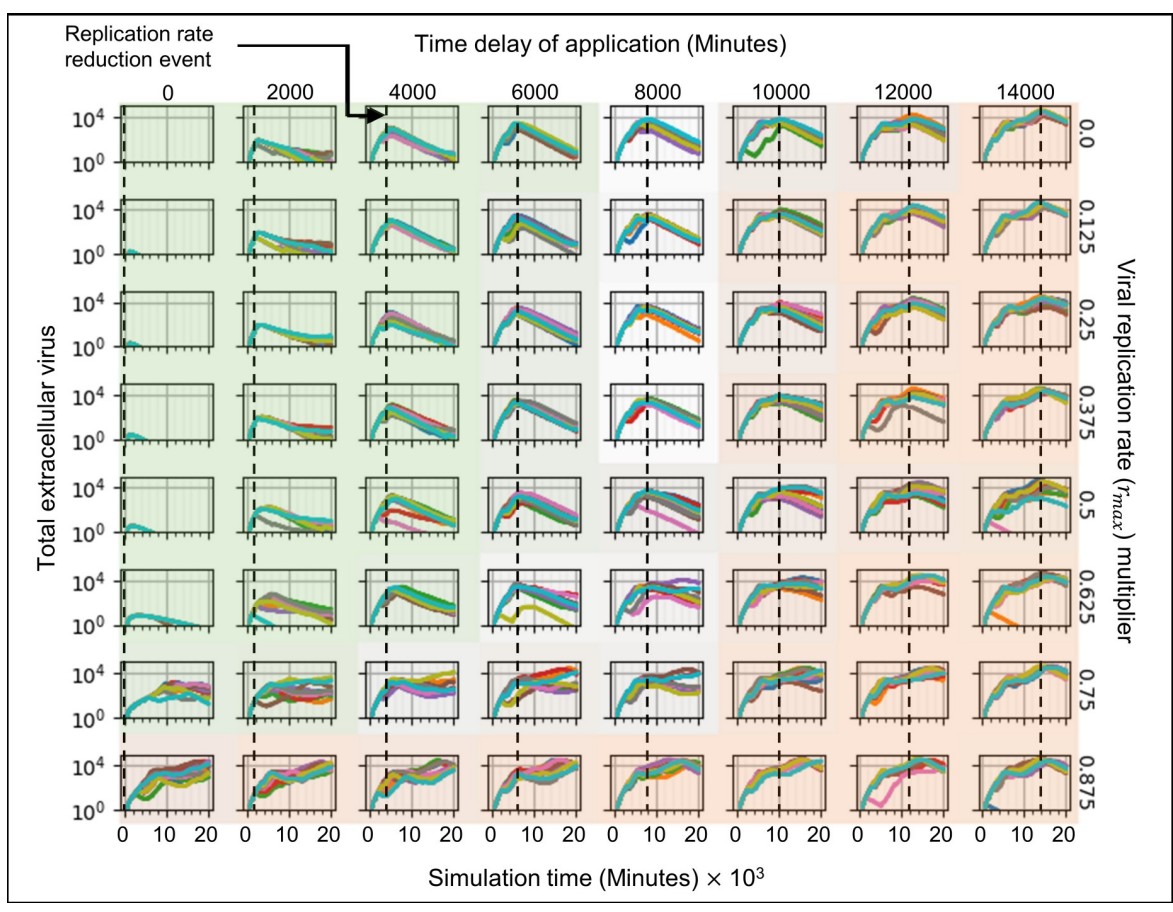

**Fig 10. Levels of extracellular virus vs time in simulations of a hypothetical drug treatment reducing the viral genome (e.g. RNA for SARS-CoV-2) replication rate ($r_{max}$) as a function of drug potency (one minus the viral replication rate multiplier) and time of initiation of treatment.** Extracellular virus is cleared or near-cleared when intervention occurs soon after infection. Parameter values, axis types and time-scale and shading as in Fig 8.

A particular parameter set (time delay of application of 10000 minutes, 100% potency) produced simulation replicas that had instances of all three outcomes (Fig 11). In a simulation replica with a positive outcome (Run 2, Fig 11A), the first uninfected epithelial cell dies (as well as a few uninfected epithelial cells) before 4000 minutes (67 hours, 2 ¾ days), after which the total extracellular virus gradually decreases. At around 4000 minutes (67 hours, 2 ¾ days), an epithelial cell near the initially-infected cell becomes infected, causing a recurrence of infection, whose spread was stopped by the treatment. In contrast, simulation replicas with intermediate and negative outcomes (Runs 8 and 4, respectively, Fig 11A) have comparable, and significantly more, numbers of infected and virus-releasing epithelial cells at 4000 minutes (67 hours, 2 ¾ days), while total extracellular virus is greater in the replicas with a negative outcome than in the replicas with an intermediate outcome. For the positive outcome replica, after 10000 minutes (167 hours, 7 days), the remaining extracellular virus infects just a few individual epithelial cells throughout the tissue. For the intermediate outcome replica, after 10000 minutes (167 hours, 7 days) the number of infected epithelial cells continues to increase until around 12000 minutes (200 hours, 8 ⅓ days) and then declines, while the number of uninfected epithelial cells slightly decreases at the end of the simulation. For the negative

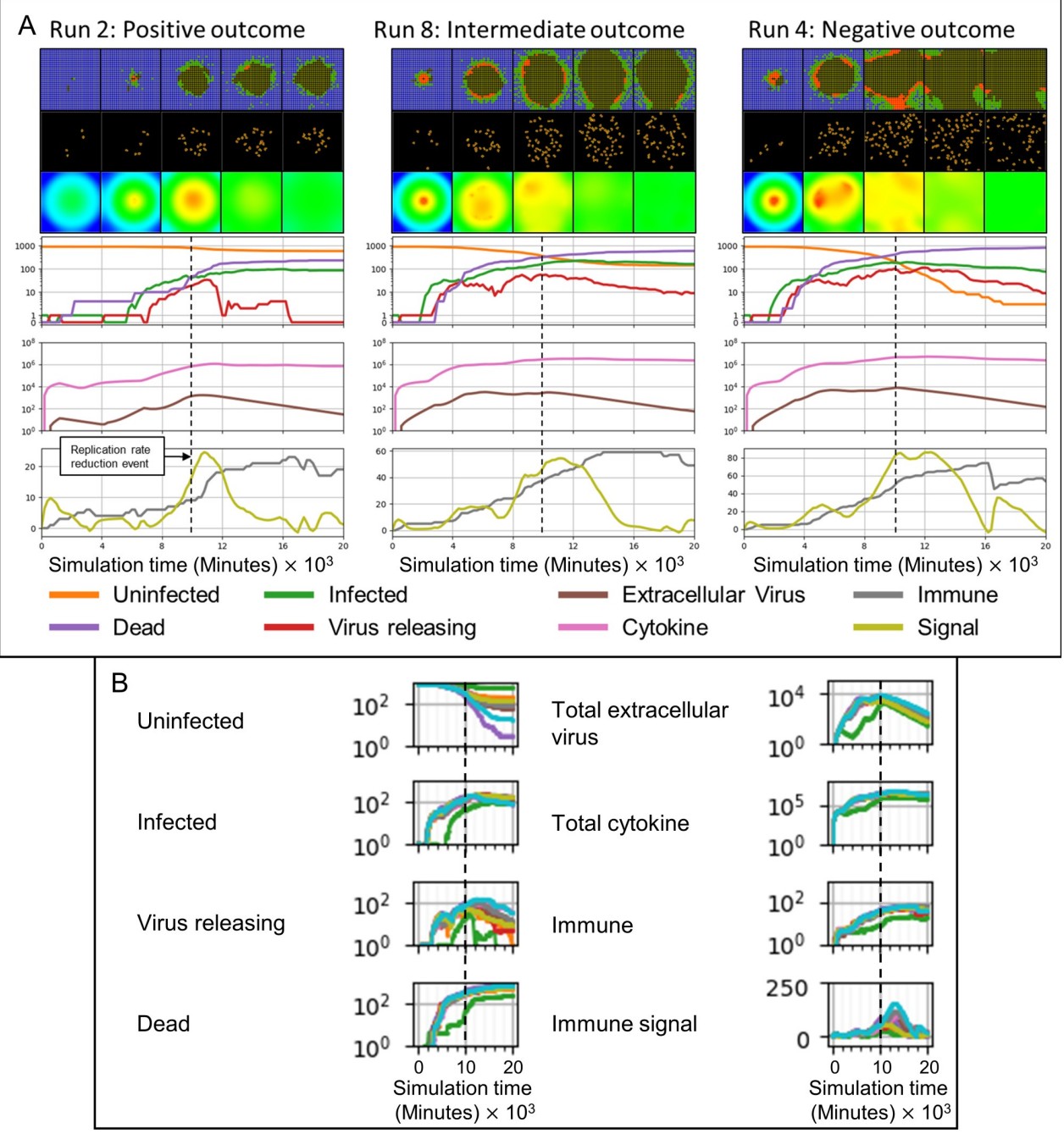

**Fig 11. Difference in treatment effectiveness for different simulation replicas for perfect treatment potency (0 viral replication rate multiplier) near the time when the extracellular virus amount would reach its maximum in the untreated case (10000 minutes, 167 hours, 7 days).** (A) Select simulation replicas for this parameter set showing the variety of possible outcomes (treatment effectiveness). Spatial results show the epithelial and immune cell layers, and the extracellular virus field, at 4000, 8000, 12000, 16000, and 20000 minutes (67, 133, 200, 267 and 333 hours, 2 ¾, 5 ½, 8 ⅓, 11 and 14 days). Cell type colors are the same as in Fig 3A. Virus field values are scaled as in Fig 3A. (B) Time series for all simulation replicas for the selected parameter set: Right column, from top to bottom, number of uninfected epithelial cells, number of infected epithelial cells, number of virus-releasing epithelial cells, number of dead cells. Left column, from top to bottom: total amount of extracellular virus, total amount of cytokine, number of immune cells and immune response state variable. All variables except the immune signal plotted on a logarithmic scale *vs* time.

outcome replica, after 10000 minutes (167 hours, 7 days) the already depleted number of uninfected epithelial cells continues to rapidly decrease to near zero.

## Model extensions

In this section we demonstrate the deployment of extensions to the framework described in *Models and methods*, which we will refer to as the "main framework" when discussing extensions in this and subsequent sections, as well as particularization of the framework to specific biological problems like a different virus. We accomplish this through integration of an existing model in the literature of hepatitis C virus (HCV), and elementary examples of adding models of heterogeneous epithelia and tissue recovery, one model of which is constructed from another. We include schematic representations where appropriate of the software implementation according to the architecture we have developed (and will continue to develop) to support broad scientific use through rapid, parallel model development, flexible model integration, and model sharing through our publicly available online repository (see *S2 Text* for an overview of basic deployment, implementation, and public distribution of extensions and *S1*–*S3 Code Snippets* for specific examples).

**Heterogeneous susceptibility inhibits spread of infection.** To demonstrate basic extensibility of the framework to model additional complexity associated with viral infection, we introduced a basic notion of heterogeneity to the epithelium of simulated scenario with the premise that not all epithelial cells can be infected (*i.e.*, some cells are unsusceptible, as in an actual heterogeneous respiratory epithelium) [22]. We implemented heterogeneous susceptibility by randomly selecting a fraction of the epithelial cell population at the beginning of simulation and setting the number of surface receptors of each to zero (*i.e.*, these cells have no surface receptors for internalization of the virus). Otherwise, all mechanisms and model parameter values used in simulations of heterogeneous susceptibility were the same as those used to generate results shown in Fig 3 (*i.e.*, all mechanisms described in *Models and methods* using the baseline parameter set). Implementation of randomly selected heterogeneous susceptibility is available in the add-on modules library and is hosted on our repository for public use with the module name "RandomSusceptibility". For each set of replicas we simulated ten replicas using the RandomSusceptibility module while varying the fraction of unsusceptible cells (like different locations in the lungs), from 10% to 50% unsusceptible in intervals of 10% (Fig 12).

For all fractions of unsusceptible cells, infection spread throughout the epithelial sheet (Fig 12A). No replica produced the exact definition of widespread infection, since all replicas had at least some remaining uninfected cells at the end of simulation. However, only eight out of fifty total replicas had a few epithelial cells that were not either uninfected or dead at the end of simulation (for infected cells, 1 replica for 10% unsusceptible, and for virus-releasing 1 replica for 10% and 20% unsusceptible, and 2 replicas for 30%, 40% and 50% unsusceptible). The distributions of infected and virus-releasing cells were notably different from all previous simulations with appreciable infection in that both subpopulations were noticeably intermixed with uninfected (presumably unsusceptible) cells and increasingly so with increasing fraction of unsusceptible cells. Most of these intermixed, uninfected cells (particularly those nearer to the initial site of infection) also died due to immune response mechanisms (*i.e.*, bystander effect and oxidative killing).

The final number of uninfected cells at the end of simulation increased with an increasing fraction of unsusceptible cells (Fig 12B). The rate of spread of infection decreased with an increasing fraction of unsusceptible cells (as observed by inspection in a rightward shift in the total number of uninfected cells). One replica for 20% unsusceptible cells exhibited

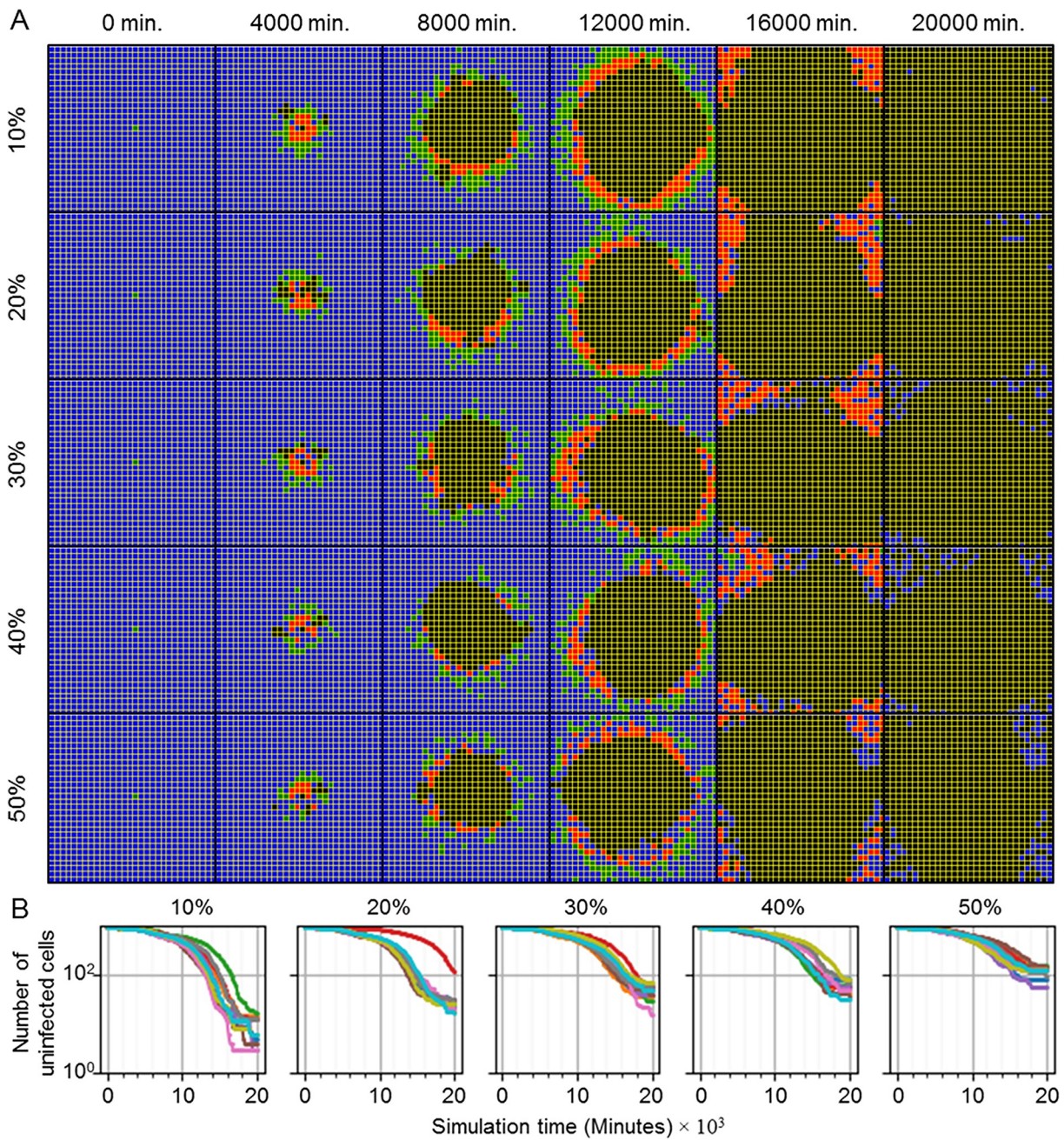

**Fig 12. Randomly distributed unsusceptible cells inhibit but do not stop spread of infection throughout the tissue.** (A) Distributions of epithelial cells during simulation time (columns) for varying fraction of unsusceptible cells (rows). From top to bottom, replicas were simulated with 10%, 20%, 30%, 40% and 50% of epithelial cells unsusceptible to viral internalization. Cell type colors are the same as in Fig 3A. (B) Number of uninfected cells during simulation time for ten replicas of each fraction of epithelial cells unsusceptible.

significantly delayed spread of infection, though this was not observed in other replicas for any other fraction of unsusceptible cells.

**Integration of an explicit RNA synthesis model allows the spatiotemporal modelling of hepatitis C virus infection.** The viral replication model described in *Models and methods* describes the viral life cycle, from internalization to release, as occurring over four stages and

in unitless quantities. As such, it is possible to integrate detailed models of various stages of the viral life cycle into the main framework through appropriate substitution of viral replication model terms and assignment of unit quantities as necessary (*i.e.*, one unit in the original viral replication model corresponds to a physical unit in an integrated model that explicitly describes a lifecycle process). Integration of a detailed model of some viral process then particularizes the virus represented in the main framework to the level of biological information introduced by the integrated model.

As a demonstration of model integration, we particularized viral replication of the framework to hepatitis C virus (HCV) using an existing model of subgenomic HCV RNA synthesis in Huh-7 cells [59]. The model of HCV RNA synthesis describes various aspects of subgenomic HCV replication like translation of HCV polyprotein in the cytoplasm, replication kinetics in vesicular-membrane structures and availability of host ribosomes, and explicitly models quantities of HCV RNA molecules, translation complexes, HCV polyprotein molecules, and necessary enzymes in both the cytoplasm and vesicular-membrane structures.

Integration of the HCV replication model casts the viral genome taking part in genome replication *R* of the viral replication module described in *Models and methods* as the cytoplasmic plus-strand HCV RNA molecules of the HCV replication model (denoted $R_p^{cyt}$, see *S1 Text*). We assumed that internalized virus from the viral internalization module converts into cytoplasmic RNA, and that some of the decay of cytoplasmic RNA molecules described in the HCV model leads to the production of quantities produced by replicating viral genome from the viral replication module. To relate the unitless viral replication model described in *E2—Viral Replication* to the biological quantities of the HCV replication model, we assumed that one unit of replicating viral genome corresponds to 100 HCV RNA molecules, which was found to produce total infection dynamics comparably to the main framework (for comparison with results of this work, rather than for reproduction *in silico* of any specific HCV data) when using both the baseline parameter set demonstrated in Fig 3 and all model parameters reported in [59] when the virus-receptor association affinity coefficient $k_{on}$ was increased by a factor of 100 (S2 Table). The integrated HCV model is implemented in the add-on modules library using the aforementioned software architecture and is hosted on our repository for public use with the module name "HCVIntegrated" (Fig 13A).

Ten simulations were executed using the integrated HCV model for a simulation time of two weeks using initial conditions similar to some used in [59], where the initially infected cell was seeded with 500 cytoplasmic viral RNA molecules (*i.e.*, initial *R* = 5, rather than initial *U* =1 as in all other simulations). Spread of infection with the integrated HCV model produced comparable patterns of spread of infection, where an infection front radially advanced outward from the initial site of infection (Fig 13B). However, one notable difference was that the prominent band of virus-releasing cells during spread of infection only occurred in some simulation replicas (Fig 13B, replica "a"), while in other replicas only small, isolated groups of cells became virus releasing (Fig 13B, replicas "b" and "c"). Variability of outcomes was particularly notable among the ten simulation replicas. Some simulation replicas produced widespread infection at a comparable timescale to that of the baseline parameter set, between one and two weeks (Fig 13C). Such simulation replicas were those that produced comparable spatial distributions of infected and virus-releasing cells (specifically, with a prominent band of virus-releasing cells) to those of the main framework using the baseline parameter set. In other simulation replicas (*e.g.*, Fig 13B, replica "c"), slowed infection occurred due to early elimination of many virus-releasing cells. In such cases, the epithelial sheet had very few virus-releasing cells and scattered infected cells around the region of dead cells.

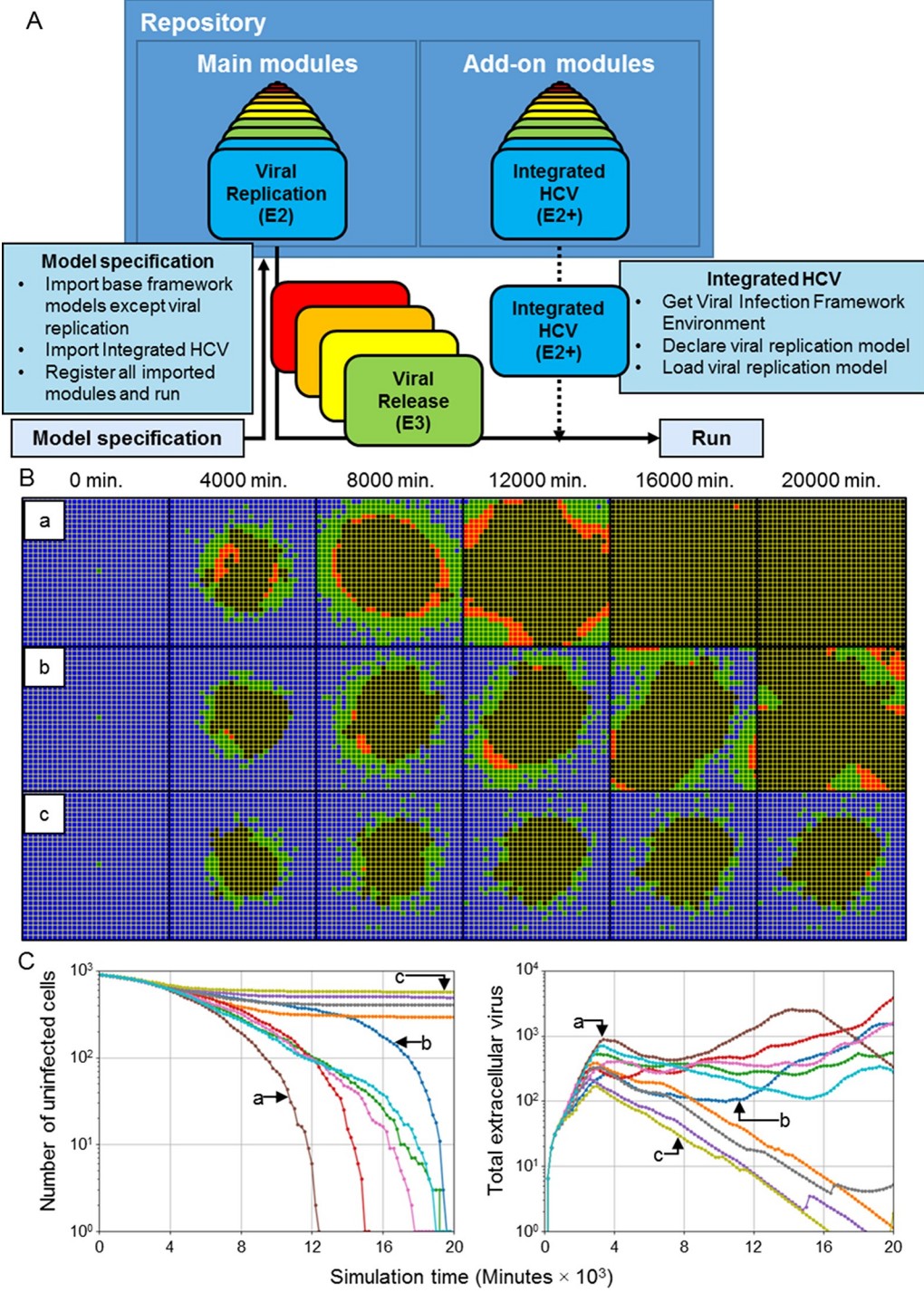

**Fig 13. Basic integration of an explicit model of hepatitis C virus subgenomic replication in a spatial context presents stochastic outcomes in ten simulation replicas.** (A) Schematic of implementing the integrated HCV model using the available modules described in *Models and methods* ("Main modules") and add-on modules libraries hosted in the framework public repository. All main modules are imported except Viral Replication, which is replaced with Integrated HCV. Modules used in a simulation are specified in model specification. The integrated HCV model module is available in the add-on modules library. (B) Distributions of epithelial cells during simulation time for select replicas (labeled "a", "b" and "c") using the integrated HCV model. Cell type colors are the same as in Fig 3A. (C) Number of uninfected cells (left) and total extracellular virus (right) during simulation time for ten replicas using the integrated HCV model. Select results shown in (B) are annotated according to replica labels "a", "b" and "c".

Many infected cells remained infected over a period of over a week, which was not observed using the viral replication model of the main framework. Such infected cells did not contribute to spread of infection, but rather diminished the likelihood of widespread infection by increasing the distance between uninfected and virus-releasing cells. By the end of simulation in replicas that produced slowed infection, total extracellular virus was negligible despite a significant number of infected cells, which presents an outcome not described in *Classification of spatio-temporal infection dynamics* that could be called "benign infection".

**An extensible framework architecture enables the inclusion of tissue recovery.** As a demonstration of modularity and extensibility, we developed, implemented and tested two models of tissue recovery, where dead cells are replaced over time with uninfected cells. Since epithelial cells in the simulated epithelial sheet are static, removal of dead cells and proliferation of uninfected cells was modeled as the changing of the type of a dead cell to uninfected (rather than explicitly modeling mitosis, [18]). To generate a scenario of viral clearance and significant tissue damage, we simulated the baseline parameter set with the parameter variations in the top-right corner of Figs 5–7, where viral internalization is severely inhibited and the immune response is very strong and fast (*i.e.*, parameters $k_{on}$ and $\beta_{delay}$ were reduced by a factor of 100 so that the virus is cleared but many uninfected cells die).

In the first model of tissue recovery, called "Simple Recovery", dead cells are replaced by an assumed layer of proliferative cells underneath the simulated epithelial patch, and so each dead cell has a fixed probability of recovering. We approximated the probability of recovery based on an assumed onset of tissue recovery of 7 days, in which case the probability of recovery for each dead cell over a 20-minute simulation step was $1.98\times10^{-3}$. In the second model of tissue recovery, called "Neighbor Recovery", dead cells are replaced by nearby uninfected cells similarly to wound healing, and so each dead cell has a probability of recovery equal to the number of neighboring uninfected cells multiplied by a coefficient. For comparison of results to those from the Simple Recovery add-on module, we used the same probability coefficient value such that the probability of recovery according to Simple Recovery for each dead cell over a 20-minute simulation step was $1.98\times10^{-3}$ per unit of contact area with neighboring uninfected cell (measured in number of neighboring lattice sites).

Both models were implemented in CompuCell3D in the add-on modules library using the aforementioned software architecture and is hosted on our repository for public use. The Simple Recovery model is hosted with the module name "RecoverySimple", and the Neighbor Recovery model is hosted with the module name "RecoveryNeighbor". Since the only difference between the two recovery models is the criterion for cell recovery (*i.e.*, whether a fixed probability, or a probability according to the neighborhood of a cell), the implementation of the Neighbor Recovery model inherits all features of the Simple Recovery model implementation using basic Python class inheritance, and required overwriting only one function that implements a criterion of cell recovery during development (Fig 14A, see *S2 Text, Extending a model in CompuCell3D*).

All simulation replicas for each recovery model began with tissue insult due to oxidative killing by a strong and fast immune response through the first few days (Fig 14B). All simulation replicas experienced a loss of approximately 100–200 uninfected cells (Fig 14C). An increasing number of uninfected cells clearly demonstrated immediate effects of modeling recovery after oxidative killing subsided (a trend not seen in previous results), where the Simple Recovery model almost completely replenished all killed uninfected cells by the end of simulation, and the Neighbor Recovery model replenished all killed uninfected cells between 8,000 and 13,000 minutes. Neighbor-dependence of the Neighbor Recovery model generated a significantly different distribution of dead cells after oxidative killing, where dead cells were scattered throughout the original region where cells died when using the Simple Recovery

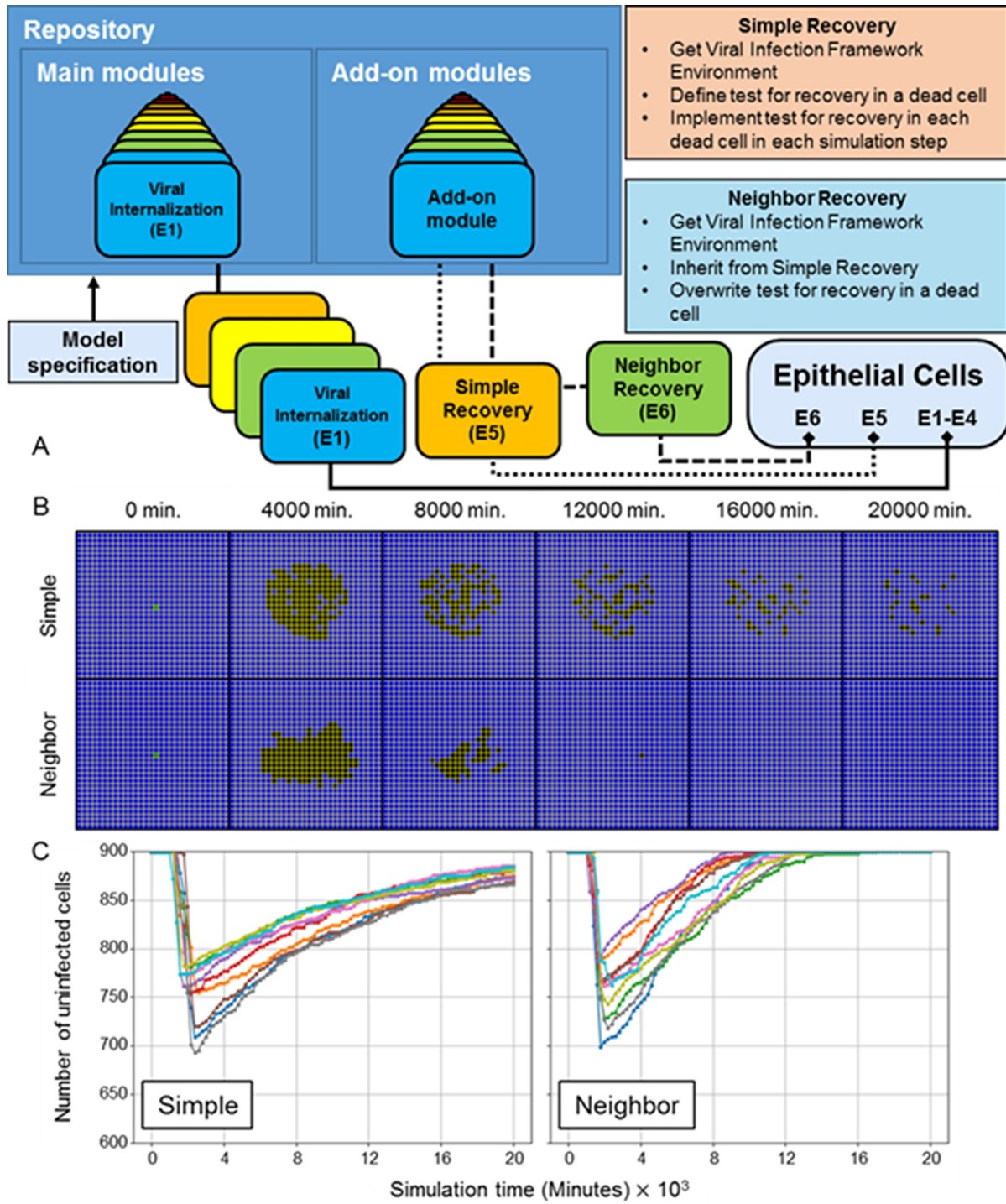

**Fig 14. Even simple model extensions and extensions of extensions can produce notably different spatiotemporal emergent features.** (A) Schematic of implementing a modified epithelial cell model using the available modules described in *Models and methods* ("Main modules") and add-on modules libraries hosted in the framework public repository. Modules used in a simulation are specified in model specification. The Simple Recovery and Neighbor Recovery model modules are available in the add-on modules library. (B) Distribution of epithelial cells during simulation time using the Simple Recovery (top) and Neighbor Recovery (bottom) models. Cell type colors are the same as in Fig 3A. (C) Number of uninfected cells during simulation time for ten simulation replicas using the Simple Recovery (left) and Neighbor Recovery (right) models.

model while the region where cells died shrank over time when using the Neighbor Recovery model.

## Discussion

Our spatial, multicellular model of primary acute viral infection of an epithelial tissue includes key aspects of viral infection, viral replication and immune response. By investigating sensitivity to model parameters and simulating drug therapies, we identified six distinct spatiotemporal classes of infection dynamics based on the model's transient behaviors and final configurations. Each of our simulation-defined classes corresponds to biologically or clinically observable factors and outcomes. The case of no immune response would be useful for analyzing *in vitro* experiments (*e.g.*, organoids). Widespread infection corresponds to an initial infection that is likely to spread to surrounding tissue and cause major tissue damage. Slowed infection corresponds to an initial infection whose spread is more likely to be eliminated by the adaptive immune response. Containment corresponds to immune-cell elimination of all infected cells but where remaining extracellular virus could result in new sites of infection elsewhere. Recurrence corresponds to the situation when new lesions form within the observed tissue patch. Clearance corresponds to immune-cell-based elimination of all infected cells and extracellular virus (classical viral clearance).

We showed that key parameters of the model, such as those affecting viral internalization (*i. e.*, virus-receptor association affinity $k_{on}$), can lead to both containment/clearance (*e.g.*, small $k_{on}$, Figs 5–7) and widespread infection (*e.g.*, large $k_{on}$, Figs 5–7). Multidimensional parameter sweeps showed how the interplay between immune response (*e.g.* immune response delay coefficient $\beta_{delay}$) and viral spread could lead to widespread infection (*e.g.*, large $\beta_{delay}$, large $k_{on}$, Figs 5–7), rapidly cleared infection (*e.g.*, small $\beta_{delay}$, small $k_{on}$, Figs 5–7) or containment/ clearance after substantial damage (*e.g.*, small $\beta_{delay}$, moderate $k_{on}$, Figs 5–7). Some of these outcomes would be expected biologically (*e.g.*, very fast internalization with a slow immune response is likely to lead to widespread infection; faster and stronger immune responses should control the spread of viral infection within the tissue [Figs 5–7]) and would also occur in deterministic non-spatial models. Others, like the coexistence of replicas with containment/clearance or failure to control for the same parameter set, are less expected, and could not occur in a deterministic non-spatial model (though they might occur in some stochastic non-spatial models). We have observed this interplay of parameters, as well as the potential for stochastic outcomes, in variations of other parameters of the model, whether related to spatial (*e.g.*, viral and cytokine diffusion coefficients) or deterministic and stochastic cellular aspects (oxidative agent threshold for death and virally-induced apoptosis dissociation coefficient, see *S12*–*S17 Figs*).

We studied the influence of timing and potency of an RNA-polymerization inhibitor like remdesivir [60] on the spread of viral infection within tissue (Figs 9–11). As expected, in our model, drugs with this mode of action can improve viral control in tissue if administered prophylactically at high potency, and their effectiveness decreases the later they are administered. Less obviously, the lower-left region of Figs 9–11 shows how therapies with even reduced potency could control the infection when administered sufficiently early, consistently with predictions from a deterministic, non-spatial ODE model (though the mode of action is not explicitly described) [61]. While we expect prophylactic or early treatment at the same potency to be more effective than later treatment, our model suggests that, for antivirals, time of treatment is a more significant factor than potency in determining the effectiveness of the therapy. Our model thus suggests that drugs that interfere with virus replication are significantly more effective if used even at very low doses prophylactically or very soon after infection, than they

would be if used even at a high dose as a treatment given later after exposure. Specifically, a prophylactic treatment in simulation which reduces the rate of viral RNA synthesis by only 35% (35% potency) is more effective than a treatment with 100% potency given two and a half days after infection, and has about the same efficacy as a treatment with 50% potency given one day after infection. Our model also showed that because of stochasticity in viral spread, later treatment at moderate to high potency may still be effective in a subset of individuals.

Both parameter sweeps had regions with little variation in outcome between replicas (*e.g.*, the upper-right and lower-left corners of Fig 5). In regions of the parameter space between these extremes (*e.g.*, the unshaded areas in Figs 5 and 9), different replicas showed dramatically different outcomes. One such parameter set in our drug therapy simulations produced three distinct qualitative outcomes (*i.e.*, positive, intermediate and negative outcomes, Fig 11). For these parameters, replica outcomes were particularly sensitive to stochasticity early in infection when only a few cells were infected (Fig 11A), with delayed spread of infection from the first infected cell producing more positive outcomes. Simulation replicas with negative outcomes (Fig 11A, Run 8) had higher extracellular virus levels at earlier times than those with interme- diate outcomes (Fig 11A, Run 4), even though the fraction of each cell type was similar. Since the viral replication module is deterministic, the primary cause of this difference is the spatial distribution of cells. Spatial structure (*e.g.*, infection of neighboring cells), stochastic events (*e. g.*, early cell death of infected cells before significant virus release) and cell-to-cell variation (*e. g.*, difference in viral release between cells) all affect the variation between replicas.

Differences in spatiotemporal dynamics and variability of outcomes thus critically depend on the ability of the model to resolve the spread of virus and immune response spatially. The intrinsic stochasticity of many model processes makes the spatial patterns of the infection front and distribution of tissue damage nontrivial. The spectrum of outcomes in our parameter sweeps (Figs 5–7 and 9–11) depends not only on parameter values and model immune response, but also on the emergent spatial patterns of cytokine and virus fields (*e.g.*, variations within the infection front expose different numbers of uninfected epithelial cells to the immune response). Such stochastic and spatial aspects can also introduce new considerations to ODE models that have been primarily employed in a non-spatial context. For example, as described in the original presentation of the HCV model that was integrated in *Integration of an explicit RNA synthesis model allows the spatiotemporal modelling of hepatitis C virus infec- tion*, the subgenomic kinetics of the HCV model require a minimum number (seven) of cyto- plasmic viral RNA molecules to reach a saturated state. When employing the HCV model in a multicellular, heterogeneous context, insufficient internalization of a spatially heterogeneous extracellular viral field for subgenomic replication to produce rampant viral production, lead- ing to insufficient cytokine signaling to invoke further immune response, makes possible the so-called outcome of benign infection.

## Future perspectives

Our modeling framework can improve with the inclusion of additional cellular and immune mechanisms discussed in Fig 1. The modularity of model modules and built-in extensibility of the publicly available software implementation enables us, and other interested members of the scientific community, to accomplish such activities collaboratively or independently, con- currently, and even when in theoretical disagreement (see *S2 Text*). Modules accounting for viral clearance, tissue recovery and persistent adaptive immune response can be added to model later stages of disease progression (as demonstrated in *Model extensions*). The current immune model does not include important signaling factors (*e.g.*, interferon-induced viral resistance in epithelial cells) and the different roles of tissue-local and systemic signals (*e.g.*,

various cytokines). It also omits many cell types associated with both innate and adaptive immune response and their roles (*e.g.*, viral scavenging by macrophages, relaying and amplification of immune signals by dendritic cells). Of special interest to results like those presented in this work is the effect of specific roles by individual immune cell phenotypes on emergent dynamics and outcomes, considering that the timing of their activities during progression of events can be quite different (*e.g.*, early neutrophil release of oxidative agent contrasted with later effector T-cell contact-mediated killing). Such details, which we are currently pursuing, are particularly important for using framework to interrogate the spatiotemporal details of the immune response, which are poorly understood. The model does not currently consider the production and role of antibodies in the humoral immune response or tissue recovery after damage (like the demonstration models presented in *An extensible framework architecture enables the inclusion of tissue recovery*). The model also greatly simplifies the structure of the epithelium and its environment, but could be easily generalized to a detailed, three-dimensional geometry, albeit at the cost of computational performance.

Our current results suggest priorities for improving the biological realism within existing modules, and for including modules representing additional biological components and mechanisms. We are currently implementing virus-scavenging by immune cells and local antiviral resistance due to Type 1-IFN paracrine signaling by epithelial cells. We are calibrating the virus replication module to existing experimental data for SARS-CoV-2 and influenza A. Because different tissues within the body have different responses to local viral infection, developing our framework to support the modeling and simulation of multi-organ disease progression (*e.g.*, by identifying model parameters corresponding to specific tissues and physiological compartments) would allow us to understand the highly variable whole-body progression of many viral diseases.

The immune response to viral infection depends on locus of infection, degree of infection and patient immune state. Understanding the reasons for immune failure to contain infection, or pathological responses like cytokine storms or sepsis, requires models of immune response at multiple locations and scales. The same is true for understanding and predicting the possible protective or adverse effects of coinfection. The number of permutations of infection timing and combination of pathogens is too large to address purely by experiments, but could be addressed by simulations. Spatial modeling is also important because the spatiotemporal dynamics of coinfection within tissues may be important to the outcome (*e.g.*, whether individual cells can be superinfected, whether viral lesions with a tissue are disjoint or overlap, whether the main foci of the pathogens are in the same or different tissues).

We can also study the systemic effects of possible therapies with known molecular modes of action (as seen in *Results*). Evaluating therapies in a simulated context prior to performing animal or human trials could lead to more effective and rapid drug discovery and to optimized dosage and timing of treatments. Understanding the origins of population variability in disease progression is crucial to providing optimal personalized treatment. While the simulations presented here begin with a single infected cell, a simulation which begins with multiple infected cells might better represent the infection dynamics of patients that have been subject to high level exposure, such as healthcare workers. Factors such as hypertension, immunosuppression and diabetes affect tissue state and immune response and could also be incorporated into our model. More detailed studies of these factors using our model could reveal more about the effects of population variability (due to age, genetic variation, prior drug treatment or immune status) on disease progression. Such computational studies could be accomplished using recently published concomitant, calibrated ODE-based simulations of COVID-19 treatment [61].

We are working to implement validated non-spatial models of viral infection and immune response as agent-based spatial models (*e.g.*, viral production, cytokine secretion, tissue damage). By starting with a validated model that uses ordinary differential equations and adding spatial components gradually, we can calibrate our spatial models and validate our results. In ongoing work, we have developed a formal method for spatializing ODE models and employing their parameters such that these analogous spatial models reproduce the ODE results in the limit of large diffusion constants. Using this method, we can combine the ability to do rapid formal parameter identification of ODE-based models and to leverage published ODE model parameters with the flexibility of spatial modeling. In these cases, any differences between the ODE results and the spatial model can be definitively attributed to spatialization (*e.g.* the local spread of virus or cytokine or the limited speed of movement of immune cells), or to additional factors which are difficult to include in an ODE model (*e.g.*, the variability of individual cells or the complex time course of virus release by individual infected cells). We have developed formal spatializations of a number of interesting ODE models of COVID, such as [61,62], to explore the effects of stochasticity of outcomes, the effects of spatial mechanisms, and infection dynamics at a particular site of infection on the predictions of these models. An additional benefit of our approach is that we can easily and consistently combine and integrate ODE models which focus on different aspects of the complex process of infection, spread and clearance (*e.g.*, combining published models of intracellular INF-induced viral resistance with spatial models of plaque spread *in vitro* [20,63]). We illustrate both of these strengths in *Integration of an explicit RNA synthesis model allows the spatiotemporal modelling of hepatitis C virus infection*, where we integrate a published HCV model of subgenomic replication into our framework. We can also conduct simultaneous, cross-platform validation of spatial models by building multiple implementations of the same conceptual and quantitative models on independent modeling platforms (here Chaste [64,65] and Morpheus [66]).

The COVID-19 crisis has shown that drug discovery and therapy development both require new predictive capabilities that improve their effectiveness and efficiency. We have developed our framework to explore the relationship between molecular, cellular-level and systemic mechanisms and outcomes of acute viral infections like SARS-CoV-2, and to support development of optimal, patient-specific treatments to combat existing and new viruses.

## Models and methods

In this section we first present our model as a high-level conceptual model where we list each process included in an implementation-independent manner. We then detail the quantitative model and its computational implementation, which uses a Cellular Potts representation of cellular dynamics. All quantitative models are implemented in a modular, extensible simulation architecture built using the CompuCell3D simulation environment, which is publicly available for download and further development by interested members of the scientific community (see *S2 Text*).

### Conceptual model: Biological hypotheses and assumptions

As discussed in *Introduction* (Fig 2A) we consider viral propagation in an epithelial tissue and a lymph node. The tissue contains two interacting spatial components: an epithelium component (consisting of a monolayer of epithelial cells), and an extracellular environment component (containing immune cells, extracellular virus and chemicals). The lymph node component (whose state is affected by signaling from the tissue) adds immune cells to the extracellular space when in a proinflammatory state and removes them when in an anti-inflammatory state. A set of processes and interactions govern how the states of these

components evolve in time. We detail these components, processes and interactions in the following subsections and in Fig 2.

**Epithelium component.** The epithelium component of the model represents the layer of epithelium in the tissue, and is composed of epithelial cells of four types: uninfected, infected, virus-releasing and dead (Fig 2C). We assume that the epithelial cells are immobile. We implicitly model the ECM by considering its influence on all processes in the epithelium component. The epithelial cells contain modules that describe the viral life cycle and approximate the amount of virus as a continuous quantity (Fig 2B), including: binding and internalization of viral particles from the extracellular environment (**E1**), intracellular replication (**E2**) and release (**E3**) of synthesized virus into the extracellular environment, as well as cell death caused by viral-replication-associated damage, immune-cell killing and oxidative agent killing (**E4**).

*E1—Viral internalization.* Module E1 models extracellular virus binding to epithelial cell receptors and internalization (including endocytosis-dependent and -independent routes). Internalization of viral particles involves binding of the viral spike protein to target cell-surface receptors, truncation by surface proteins and receptor-mediated endocytosis or fusion with the host plasma membrane. We assume the dynamics of internalization can be represented by describing the dynamics of virus-surface-receptor binding, determined by the amount of extracellular virus and target surface receptors, and by the binding affinity between them (**T1→E1**). We also consider the dynamic depletion of unbound target surface receptors on a cell when it internalizes a virus and superinfection of infected cells. Internalized viral particles initiate the viral replication process (**E1→E2** and Fig 2B).

*E2—Viral replication.* Module E2 models the viral replication cycle inside a host epithelial cell (Fig 2B). Individual cells infected with many non-lytic viruses show a characteristic three-phasic pattern in their rate of viral release. After infection and during an *eclipse phase*, a cell accumulates but does not yet release newly assembled viruses. In a second phase, the rate of viral release increases exponentially until the virus-releasing cell either dies or, in a third phase, saturates its rate of virus synthesis and release. Viral replication hijacks host synthesis pathways and may be limited by the availability of resources (amino acids, ATP, *etc.*), synthesis capability (ribosomes, endoplasmic reticulum, *etc.*) or intracellular viral suppression. A quantitative model of viral replication needs to be constructed and parameterized such that it reproduces these three phases.

We model viral replication based on processes associated with positive sense single-stranded RNA (+ssRNA) viruses. +ssRNA viruses initiate replication after unpacking of the viral genetic material and proteins into the cytosol (**E1→E2**). The viral RNA-dependent RNA polymerase transcribes a negative RNA strand from the positive RNA strand, which is used as a template to produce more RNA strands (denoted by "Viral Genome Replication" in Fig 2B). Replication of the viral genome is the only exponential amplification step in the growth of most viruses within cells. Subgenomic sequences are then translated to produce viral proteins ("Protein Synthesis" Fig 2B). Positive RNA strands and viral proteins are transported to the endoplasmic reticulum (ER) where they are packaged for release. After replication, newly synthesized viral genetic material is translated into new capsid protein and assembled into new viral particles ("Assembly and Packaging" in Fig 2B). These newly assembled viral particles initiate the viral release process (**E2→E3**). We assume the viral replication cycle can be modeled by defining four stages: unpacking, viral genome replication, protein synthesis, and assembly and packaging. Fig 2B illustrates these subprocesses of replication and their relation to viral internalization and release.

*E3—Viral release.* Module E3 models intracellular transport of newly assembled virions and release into the extracellular environment (**E3→T1** and Fig 2B "Release"). We conceptualize the virus being released into the extracellular fluid above the apical surfaces of epithelial cells.

Newly assembled virions are packed into vesicles and transported to the cell membrane for release into the extracellular environment (**E2→E3**). We assume that no regulation occurs after assembly of new virus particles, and that release into the extracellular environment can be modeled as a single-step process (**E3→T1**).

*E4—Cell death*. Module E4 models death of epithelial cells due to various mechanisms. Models the combined effect of the many types of virus-induced cell death (*e.g.*, production of viral proteins interferes with the host cell's metabolic, regulatory and delivery pathways) as occurring due to a high number of assembled viral particles in the viral replication cycle (**E2→E4**). Models cell death due to contact cytotoxicity (**I3→E4**). Models cell death due to oxidizing cytotoxicity (**T3→E4**).

**Extracellular environment component.** The extracellular environment contains the immune cells, extracellular virus, cytokines and oxidative agent, and is the space where transport of viral particles (**T1**), cytokine molecules (**T2**) and the oxidizing agent (**T3**) occurs. We implicitly model the ECM in the extracellular environment by subsuming its geometrical, biochemical and biophysical influences on immune cell motility and virus/cytokine/agent spreading in the chosen rate laws and parameter set. Immune cells are mobile and can be either activated or inactive (**I1**). Inactive immune cells move through random cell motility and activated immune cells chemotax along the cytokine field (**I2**). The immune cell modules also account for cytotoxic effects of immune cells on contact due to antigen recognition (**I3**) and through the secretion of oxidizing agents (**I4**).

*T1—Viral transport*. Module T1 models diffusion of viral particles in the extracellular environment and their decay. Viral particles are transported by different mechanisms (*e.g.*, ciliated active transport, diffusion) and media (*e.g.*, air, mucus) at different physiological locations and through different types of tissue (*e.g.*, nasopharyngeal track, lung bronchi and alveoli). Viral particles are eliminated by a variety of biological mechanisms. We represent these mechanisms by modeling transport of viral particles as a diffusive virus field with decay in the extracellular environment. We model transport in a thin layer above the apical surfaces of epithelial cells. Viral internalization results in the transport of a finite amount of virus from the extracellular environment into a cell and depends on the amount of local extracellular virus and number of cell surface receptors (**T1-E1**). Infected cells release viral particles into the extracellular environment as a result of the viral replication cycle (**E3-T1**).

*T2—Cytokine transport*. Module T2 models diffusion and clearance of immune signaling molecules in the extracellular environment. The immune response involves multiple signaling molecules acting upon different signaling pathways. We assume that the complexity of immune signaling can be functionally represented using a single chemical field that diffuses and decays in the extracellular environment. Once infected, epithelial cells secrete signaling molecules to alert the immune system (**E2-T2**). Locally, exposure to cytokine signaling results in activation of immune cells (**T2-I1**). Upon activation, immune cells migrate towards infection sites guided by cytokine (**T2-I2**). Lastly, activated immune cells amplify the immune signal by secreting additional cytokines into the extracellular environment (**I1-T2**). We model long-range effects by assuming that cytokine exfiltrates tissues and is transported to immune recruitment sites (**T2-L1**).

*T3—Oxidizing agent burst and transport*. Module T3 models diffusion and clearance of a general oxidizing agent in the extracellular environment. One of the cytotoxic mechanisms of immune cells is the release of different oxidizing agents, reactive oxygen species like $H_2O_2$ and nitric oxide. The mechanism of action of such agents varies but we assume that we can generalize such effects by modeling a single diffusive and decaying oxidizing agent field in the extracellular environment. The oxidizing agent is secreted by activated immune cells after persistent exposure to cytokine signals (**I4→T3**). We assume that the range of action of the

oxidizing agent is short. Cell death is induced in uninfected, infected and virus-releasing epithelial cells when sufficiently exposed to the oxidizing agent (**T3→E4**).

*I1—Immune cell activation*. Module I1 models immune cell maturation due to cytokine signaling. Immune cells mature at the recruitment site before being transported to the infection site as inactive immune cells (**L1→Immune Cells**). After infiltration, immune cells need to be exposed to local cytokine signals before activating (**T2→I1**). Once activated, immune cells chemotax along the cytokine field (**I2**) and amplify immune signaling by releasing cytokine molecules into the extracellular environment (**I1→T2**). Immune cells can also deactivate after a period of activation (**I1** and Fig 2C).

*I2—Immune cell chemotaxis*. Module I2 models activated immune cell chemotactic migration towards infection sites. We assume that upon activation (**I1→I2**), immune cells move preferentially towards higher concentrations of local cytokine (**T2→I2**).

*I3—Immune cell direct cytotoxicity and bystander effect*. Module I3 models immune cell cytotoxicity when immune cells (both activated and inactive) identify and induce death in epithelial cells with internal virus. Immune cells identify epithelial cells with internal virus on contact by antigen recognition and induce cell death by activating the caspase cascade (**I3→E4**). Uninfected, infected, and virus-releasing epithelial cells in contact with an epithelial cell that is killed by direct cytotoxicity can die through a bystander effect.

*I4—Immune cell oxidizing agent cytotoxicity*. Module I4 models activated immune cell killing of target cells through the release of a diffusive and decaying oxidizing agent into the environment. Cell death is induced in uninfected, infected and virus-releasing epithelial cells when sufficiently exposed to the oxidizing agent (**T3→E4**).

**Lymph node component.** The lymph node component models the net pro- or anti-inflammatory state of the immune system. It responds to cytokines received from the tissue and adds or removes immune cells from the tissue (**L1**).

*L1—Immune cell recruitment*. Module L1 models immune cell recruitment and infiltration into the tissue in response to cytokine signaling by infected cells and activated immune cells. Infected cells secrete signaling molecules into the extracellular environment (**E2→T3**), which alerts resident immune cells and recruits new immune cells from the blood, distant lymph nodes and bone marrow. We model the local strength of the cytokine signal as causing an increase in the strength of the signal at the immune recruiting sites. We model long-distance signaling by assuming that cytokine molecules in the extracellular environment exfiltrate the infection site and are transported through the lymphatic system to the lymphatic system to lymph nodes and through the bloodstream to initiate immune-cell recruitment (**T2→L1**). A delay on the order of minutes to hours would represent semi-local recruitment (*e.g.*, at the blood vessels). A delay on the order of days would represent long-range, systemic recruitment (*e.g.*, the time required for a dendritic cell to reach a lymph node and an induced T cell to return). Recruited immune cells are then transported and infiltrate the infection site (**L1→Immune Cell**).

## Quantitative model and implementation

For model construction and integration we use the open-source multicellular modeling environment CompuCell3D (www.compucell3d.org) which allows rapid and compact specification of cells, diffusing fields and biochemical networks using Python and the Antimony language [55,67]. CompuCell3D is specifically designed to separate model specification (conceptual and quantitative models) from the details of model implementation as a simulation and to make simulation specification accessible to biologists and others not specializing in software development. In this paper, we specifically designed the Python modules and their cross-

scale integration to have clear and stable APIs, allowing modules to be rapidly swapped out by collaborating developers. CompuCell3D runs on Windows, Mac and Linux platforms without change of model specification, and allows cluster execution for parameter exploration.

**Cellular Potts model (CPM).** *Cell types*. Cells are divided into two broad groups, epithelial and immune cells, and have a *type* (see Fig 2C) which determines their properties, the processes and interactions in which they participate, and their events and dynamics. Epithelial cells can have one of four types (uninfected, infected, virus releasing and dead). Immune cells can have one of two types (activated and inactive). Cell types can change according to outcomes of various modules, and a module specifying such an event describes both the initial and final cell types of the transition. A cell type in the model is not a phenotype in the biological sense (*e.g.*, *epithelial* cell), but an identifier for the various states that a particular cell can assume (*e.g.*, *dead epithelial* cell). When an epithelial cell changes to the dead type, all epithelial modules are disabled and the cell is generally inactive.

*Cellular dynamics*. Cellular spatial dynamics is modeled using the Cellular Potts model (also known as *CPM*, or Glazier-Graner-Hogeweg model), which represents generalized cells and medium as occupying a set of sites in a lattice [68]. Random cell motility is modeled as the stochastic exchange of sites at intercellular and cell-medium interfaces. Configurations evolve to minimize the system's effective energy $\mathcal{H}$,

$$\mathcal{H} = \sum_\sigma \lambda_{volume}(v(\sigma) - V(\tau(\sigma)))^2 + \sum_x \sum_{x\prime \in N(x)}(1 - \delta_{\sigma(x),\sigma(x\prime)})J(\tau(\sigma(x)), \tau(\sigma(x\prime)))$$
$$+ \mathcal{H}_{chemotaxis}. \tag{1}$$

Here $\sigma$ is the integer identification of a cell and $\tau(\sigma)$ is the type of cell $\sigma$. $v(\sigma)$ and $V(\sigma)$ are the current and target volumes of cell $\sigma$, respectively, and $\lambda_{volume}$ is a volume constraint coefficient. $N(x)$ is the neighborhood of site $x$, $\delta_{i,j}$ is the Kronecker-delta, and $J(\tau, \tau')$ is the effective contact energy per unit surface area between cells of types $\tau$ and $\tau'$. The final term, $\mathcal{H}_{chemotaxis}$, models chemotaxis-directed cell motility, and is prescribed by module I2. The cell configuration evolves through asynchronous lattice-site copy attempts. A lattice-site copy attempt starts by random selection of a site $x$ in the lattice as a target, and a site $x'$ in its neighborhood as a source. A configuration update is then proposed in which the value $x'$ from the source site overwrites the value of $x$ in the target site. The change in total effective energy $\Delta\mathcal{H}$ due to the copy attempt is calculated, and the update is executed with a probability given by a Boltzmann acceptance function,

$$\Pr(\sigma(x) \to \sigma(x\prime)) = e^{-\max\left\{0,\frac{\Delta\mathcal{H}}{\mathcal{H}^*}\right\}}. \tag{2}$$

Here the intrinsic random motility $\mathcal{H}^*$ controls the stochasticity of accepted copy attempts. Updates that reduce the system's effective energy are always accepted. The unit of simulation time is the Monte Carlo step (*MCS*)–taken to be 20 minutes in this work. One MCS corresponds to considering a number of copy attempts equal to the number of lattice sites.

**Epithelial component modules.** Processes **E1**-**E4** describe epithelial cell functions as defined below. **E1**, **E2** and **E4** govern the cell-type transitions of epithelial cells (see Fig 15). **E1** transforms an uninfected epithelial cell into an infected epithelial cell. **E2** transforms an infected epithelial cell into a virus-releasing epithelial cell. **E4** transforms a virus-releasing epithelial cell into a dead cell.

*E1—Viral internalization*. To capture the stochasticity associated with internalization of discrete virus particles in terms of discrete binding events, we assign each uninfected, infected and virus-releasing epithelial cell a probability of absorbing diffusive viral particles from the extracellular virus field (**T1**). The uptake probability $\Pr(Uptake(\sigma){>}0)$ for each cell $\sigma$ is given

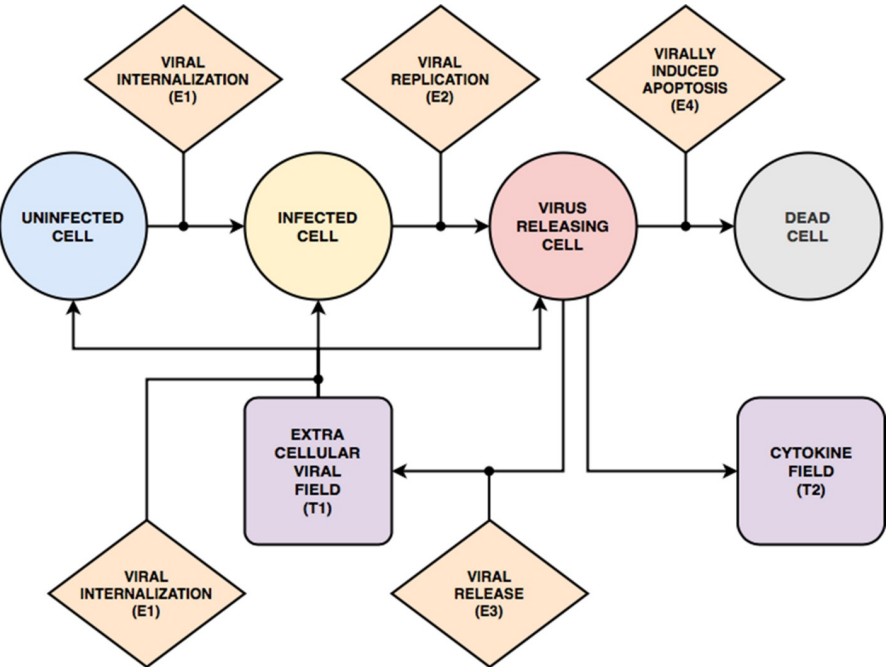

**Fig 15. State diagram and interactions of epithelial cells.Epithelial cells can have one of four "cell types": uninfected, infected, virus-releasing and dead.** Uninfected cells become infected cells when the viral uptake model (**E1**) internalizes viruses from the extracellular virus field (**T1**). Infected cells continue internalizing viruses from the extracellular virus field and become virus-releasing cells when the viral replication model (**E2**) produces sufficient newly assembled virions. Virus-releasing cells secrete viruses into the extracellular virus field (**T1**) according to the viral release module (**E3**) and secrete cytokines directly into the extracellular cytokine field (**T2**). Virus-releasing cells can die if the conditions of the virally induced cell-death model (**E4**) are met.

by a Hill equation of the total amount of diffusive viral particles in the domain of the cell $c_{vir}(\sigma)$, the number of unbound cell surface receptors $SR(\sigma)$ and the binding affinity between them.

$$\Pr(Uptake(\sigma) > 0) = \frac{\Delta t}{\alpha_{upt}} \frac{(c_{vir}(\sigma))^{h_{upt}}}{(c_{vir}(\sigma))^{h_{upt}} + V_{upt}^{h_{upt}}}, \text{ where } V_{upt} = \frac{R_o k_{off}}{2k_{on}v(\sigma)SR(\sigma)}. \qquad (3)$$

Here $h_{upt}$ is a Hill coefficient, $R_o$ is the cell's initial number of unbound receptors, $k_{on}$ is the virus-receptor association affinity, $k_{off}$ is the virus-receptor dissociation affinity, $\alpha_{upt}$ is a characteristic time constant of uptake and $\Delta t$ is the time represented by one MCS. At each simulation time step, the uptake probability is evaluated against a uniformly-distributed random variable. When uptake occurs, the uptake rate is proportional to the local amount in the virus field (**T1**), and the probability of uptake is used to define the amount (*Uptake*) of virus taken up during the MCS,

$$Uptake(\sigma) = \frac{1}{\Delta t} \Pr(Uptake(\sigma) > 0)c_{vir}(\sigma), \qquad (4)$$

$$\frac{dSR(\sigma)}{dt} = -Uptake(\sigma). \qquad (5)$$

The amount absorbed by each cell (*Uptake*) is uniformly subtracted from the virus field over the cell's domain and the cell's number of cell surface receptors is reduced by the same

amount. The amount of virus taken up (*Uptake*) is also passed to the cell's instance of the viral replication model (**E2**). Infected epithelial cells continue taking up viral particles from the environment until their cell surface receptors are depleted.

*E2—Viral replication.* Our viral replication model combines equations and parameters from several sources to represent the replication of a generic virus [7,9,52,53]. The model contains four variables representing viral quantities in different states of the viral replication process: internalized virus $U$ (Eq (6), the process of unpacking), viral genome taking part in genomic replication $R$ (Eq (7), the process of viral genome replication), synthesized protein $P$ (Eq (8), the process of protein synthesis), and assembled and packaged virions $A$ (Eq (9), the process of assembly and packaging). Biologically, the only process which can exponentially increase the rate of virus production by a single cell is viral genome replication, so the equations include the positive feedback by $R$ in Eq (7). Biologically, factors like the cell's metabolism, limited number of ribosomes, maximum rate of endoplasmic reticulum function and activation of intracellular viral suppression pathways all limit production of viral components, so we include a Michaelis-type saturation term for the rate of replication in Eq (7). Each uninfected, infected and virus-releasing cell in the simulation contains an independent copy of the system of ordinary differential equations modeling the viral replication process,

$$\frac{dU}{dt} = Uptake - r_u U, \tag{6}$$

$$\frac{dR}{dt} = r_u U + r_{max} R \frac{r_{half}}{R + r_{half}} - r_t R, \tag{7}$$

$$\frac{dP}{dt} = r_t R - r_p P, \tag{8}$$

$$\frac{dA}{dt} = r_p P - Release. \tag{9}$$

Here $r_u$ is the unpacking rate, $r_{max}$ is the viral replication rate, $r_t$ is the translating rate (rate at which viral genomes turn into RNA templates for protein production) and $r_p$ is the packaging rate. *Uptake* is defined in **E1** and *Release* is defined in **E3**. The saturation of the rate of viral genome replication is represented by a Michaelis-Menten function, $\frac{r_{half}}{R+r_{half}}$, where $r_{half}$ is the amount of $R$ at which the viral genome replication rate is reduced to $\frac{r_{max}}{2}$ (and the flux is reduced to $\frac{1}{2} r_{max} r_{half}$). The duration of the eclipse phase of single-cell infection (the time between the first entry of the virus into the cell and the first release of newly synthesized virus) is approximately $t_{eclipse} \approx \frac{1}{r_u} + \frac{1}{r_{max}} + \frac{1}{r_t} + \frac{1}{r_p}$ (11.7 hours for the reference parameter set in Table 1), with the additional complication that in our model, an epithelial cell does not release virus until $A$ reaches a threshold value. The timescale for tenfold increase of virus release when viral replication is maximal is $t_{10} \approx \frac{log(10)}{r_{max}}$ (7.7 hours for the reference parameter set in Table 1). The number of newly assembled virions is passed to the cell's instance of the viral release module (**E3**). See Fig 2B for a schematic of the viral replication process and Fig 16 for a representative time series from the viral internalization, replication and release modules.

*E3—Viral release.* Virus releasing cells release viral particles into the extracellular virus field (**T1**). The amount of virus released by a cell per unit time is proportional to the state variable

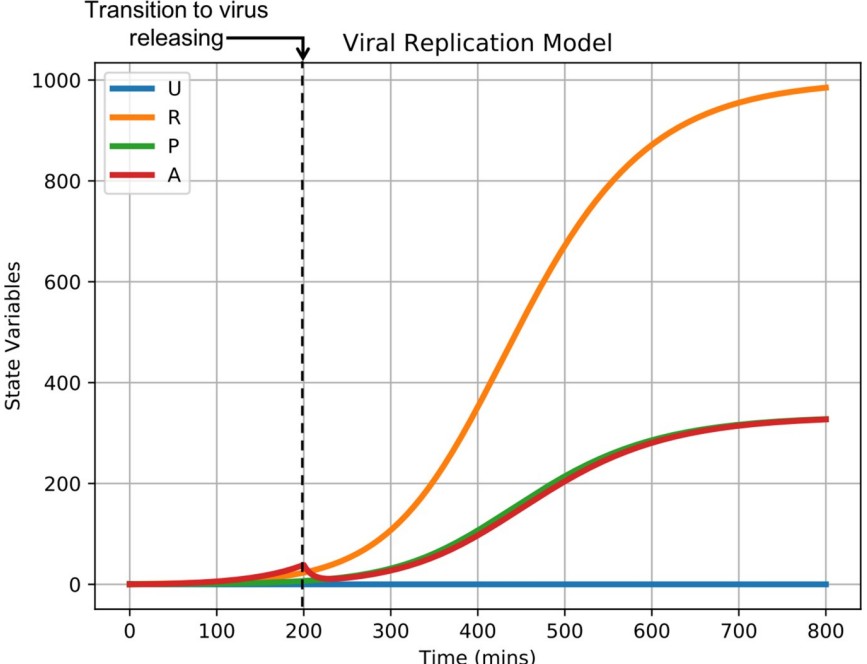

**Fig 16. Representative time series of viral internalization, replication and release models from Fig 2B.** A sample simulation of the viral replication model in a single epithelial cell. The model is initialized with one unit of internalized virus (U = 1), and the rest of the state variables set to zero (R = 0, P = 0, A = 0). No additional virus internalization occurs during this sample simulation. Dashed line indicates the time of the cell's transition from the infected to virus-releasing cell type.

for assembled virions in the viral replication module (**E2**),

$$Release = r_s A. \tag{10}$$

Here $r_s$ is the release rate of viral particles and $A$ is the level of assembled virus in the cell (defined in **E2**). The total amount released by each cell $r_s A \Delta t$ is subtracted from the cell's state variable for assembled virions $A$ and passed to the source term of the extracellular virus field (**T1**) to maintain mass balance.

*E4—Cell death*. For cell death due to virally-induced apoptosis, each infected and virus-releasing cell can die due to the amount of intracellular virus. The rate of death is defined as a stochastic function of the state variable for assembled new virions from the viral replication module (**E2**). If a virus releasing cell dies then it changes its cell type to dead and the cell's instances of the viral internalization, replication and release modules are disabled. The probability of virus-induced apoptosis per simulation step is a Hill equation of the current load of assembled virus,

$$\Pr(\tau(\sigma) \rightarrow \text{Dead} | \tau(\sigma) = \text{Virus releasing}) = \frac{\Delta t}{\alpha_{apo}} \frac{(A(\sigma))^{h_{apo}}}{(A(\sigma))^{h_{apo}} + V_{apo}^{h_{apo}}}. \tag{11}$$

where $A(\sigma)$ is the number of assembled virions in cell $\sigma$, $h_{apo}$ is a Hill coefficient, $V_{apo}$ is the amount of assembled virions at which the apoptosis probability is 0.5 per unit time and $\alpha_{apo}$ is a characteristic time constant of virally-induced apoptosis. For modeling of cell death due to contact cytotoxicity, see *I3—Immune cell direct cytotoxicity and bystander effect*. For modeling of cell death due to oxidizing cytotoxicity, see *I4—Immune cell oxidizing agent cytotoxicity*.

Regardless of the death mechanism the internally assembled virions are not released to the environment and do not take action in further infection. We assume that in the process of death the assembled virus and viral particles are either damaged or deactivated.

**Lymph node modules.** *L1—Immune cell recruitment.* The total immune cell population is governed by an ordinary differential equation of a dimensionless state variable $S$ that represents immune response due to local conditions and long-distance signaling. Our convention is that when $S>0$, immune cells are recruited to the simulation domain; likewise, immune cells are removed from the simulation domain when $S<0$. Probability functions of $S$ describe the likelihood of immune cell seeding and removal,

$$\Pr(\text{add immune cell}) = \text{erf}(\alpha_{immune}S), S > 0, \quad (12)$$

$$\Pr(\text{remove immune cell}) = \text{erf}(-\alpha_{immune}S), S < 0. \quad (13)$$

Here the coefficient $\alpha_{immune}$ is the sensitivity of immune cell addition and removal to the state variable $S$. The dynamics of $S$ are cast such that, in a homeostatic condition, a typical number of immune cells can be found in the simulation domain, and production of cytokine (**T2**) results in additional recruitment via long-distance signaling (*i.e.*, with some delay). We model this homeostatic feature using the feedback mechanism of the total number of immune cells $N_{immune}$ in the simulation domain. Cytokine signaling is modeled as perturbing the homeostatic state using the term $\alpha_{sig}\delta$. Here $\delta$ is the total amount of decayed cytokine in the simulation domain and $\alpha_{sig}>0$ models signaling by transmission of cytokine to some far-away source of immune cells. We write the rate of change of $S$ as

$$\frac{dS}{dt} = \beta_{add} - \beta_{sub}N_{immune} + \frac{\alpha_{sig}}{\beta_{delay}}\delta - \beta_{decay}S. \quad (14)$$

Here $\beta_{add}$ and $\beta_{sub}$ control the number of immune cells in the simulation domain under homeostatic conditions. $\beta_{delay}$ controls the delay between transmission of the cytokine to the lymph node and corresponding immune response by adjusting the rate of recruitment due to total cytokine (*i.e.*, increasing $\beta_{delay}$ increases the resulting delay). $\beta_{decay}$ regulates the return of $S$ to an unperturbed state (*i.e.*, $S = 0$, increasing $\beta_{decay}$ increases the rate of return to $S = 0$). To determine the seeding location, the simulation space is randomly sampled $n_{seeding}$ times, and an immune cell is seeded at the unoccupied location with the highest amount of the virus field. If no location is unoccupied, then the immune cell is not seeded. The removal probability is evaluated for each immune cell at each simulation step. Immune cells are removed by setting their volume constraint to zero.

**Immune cell modules.** The four processes **I1**-**I4** capture immune cell functions which are defined below. These processes control how immune cells are activated, translocate, and kill other cells. Their interactions with epithelial cells and other model components are illustrated in Fig 17.

*I1—Immune cell activation.* Inactive immune cells become activated with a probability according to a Hill equation of the total cytokine bound to the cell $B_{cyt}(\sigma,t)$,

$$\Pr(\tau(\sigma, t) \rightarrow \text{activated immune}|\tau(\sigma, t) = \text{inactive immune})$$

$$= \frac{(B_{cyt}(\sigma, t))^{h_{act}}}{(B_{cyt}(\sigma, t))^{h_{act}} + (EC50_{cytact})^{h_{act}}}. \quad (15)$$

After ten hours, an activated immune cell becomes inactive, in which case evaluations of activation (Eq (15)) recommence. The immune cells "forget" a percentage $(1-\rho_{cyt})$ of the

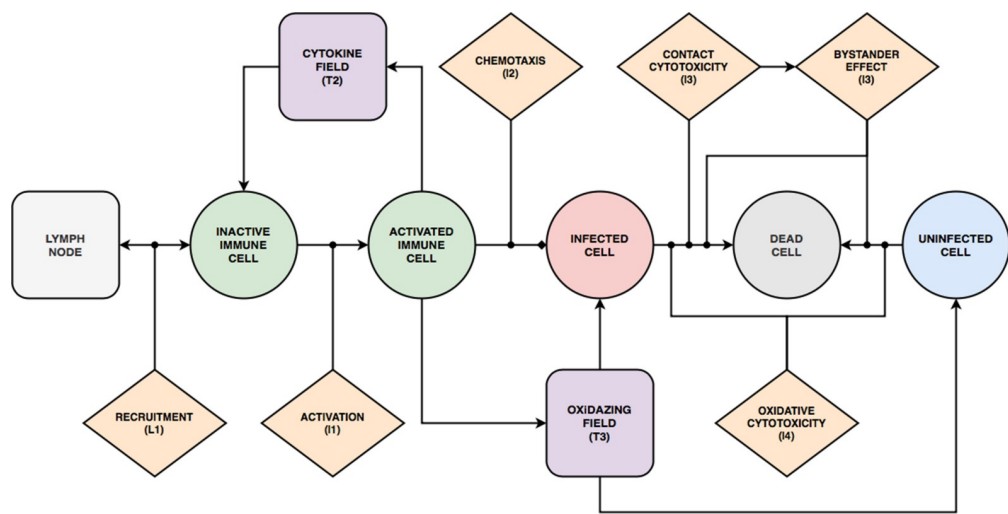

**Fig 17. State diagram and interactions of Immune cells.** Immune cells can adopt two different generalized types: inactive and activated. Inactive immune cells are recruited by the cytokine levels according to the immune recruitment module (**L1**).Transition from inactive to activated immune cells is determined by the immune activation module (**I1**) when cells are exposed to cytokines in the tissue. Activated immune cells amplify the cytokine signal by secreting cytokines to the extracellular environment. Activated immune cells chemotax towards virus-releasing cells (**I2**). Immune cells induce death of epithelial cells by direct cytotoxicity when coming into contact with infected cells (**I3**), bystander effect by killing neighbors of infected cells (**I3**) and through oxidative cytotoxicity (**I4**) by releasing cytotoxic oxidizing agents (**T3**) into the extracellular environment.

bound cytokine each time step while taking up an amount of cytokine from the environment ($\omega_{cyt}(\tau(\sigma)$,$t)$ defined in **T2**),

$$B_{cyt}(\sigma, t) = \rho_{cyt}B_{cyt}(\sigma, t - \Delta t) + \omega_{cyt}(\tau(\sigma(x)), t). \tag{16}$$

*I2—Immune cell chemotaxis.* Activated immune cells experience a motile force as a response to a signaling field. Immune cells chemotax along concentration gradients of the cytokine field. The chemotactic effective energy $\mathcal{H}_{chemotaxis}$ associated with the gradient is calculated according to a chemotactic sensitivity parameter $\lambda_{chemotaxis}$ and calculated chemotactic force $F_{chemotaxis}$. The contribution of $\mathcal{H}_{chemotaxis}$ to the change in the system's total effective energy is calculated using $F_{chemotaxis}$ when considering copy attempts. The chemotactic force at a location $x$ is saturated by normalizing the chemotactic sensitivity parameter by the concentration of cytokine at the center of mass of the cell at $x$, $c_{cyt,CM}(\sigma(x))$,

$$F_{chemotaxis}(x) = \frac{\lambda_{chemotaxis}}{1 + c_{cyt,CM}(\sigma(x))} \nabla c_{cyt}(x). \tag{17}$$

*I3—Immune cell direct cytotoxicity and bystander effect.* Immune cells, whether activated or not, have the ability to kill infected cells by direct contact. At each simulation step, immune cells trigger cell death in the infected and virus-releasing epithelial cells with which they come in contact. When an infected cell is killed by direct cytotoxicity, each of its first-order neighbors is evaluated for cell death by a bystander effect model. Each of those neighbors $\sigma' \in N(\sigma)$ in the first-order neighborhood $N(\sigma)$ of a cell $\sigma$ killed by direct cytotoxicity has a probability $k_{bystander}$ of dying from the bystander effect given by,

$$\Pr(\tau(\sigma', t) \rightarrow \text{Dead}|Direct\ Cytotoxicity(\sigma) = \text{True}) = k_{bystander} \forall \sigma' \in N(\sigma). \tag{18}$$

*I4—Immune cell oxidizing agent cytotoxicity*. Immune cells release a short-range, diffusive oxidizing agent when exposed to high cytokine concentration (**T3**). The oxidizing agent kills an epithelial cell of any type when the total amount of oxidizing agent in the domain of the cell $c_{oxi}(\sigma)$ exceeds a threshold for death $\tau_{oxi}^{death}$,

$$\Pr(\tau(\sigma, t) \rightarrow \text{Dead}|c_{oxi}(\sigma) > \tau_{oxi}^{death}) = 1. \tag{19}$$

**Extracellular environment modules.** *T1—Viral transport*. The change in concentration of the virus field $c_{vir}$ is calculated at each location in the simulation domain by solving the following reaction-diffusion equation,

$$\frac{\partial c_{vir}(x)}{\partial t} = D_{vir} \nabla^2 c_{vir}(x) - \gamma_{vir} c_{vir}(x) + \frac{1}{v(\sigma(x))} (Release(\sigma(x)) - Uptake(\sigma(x))). \tag{20}$$

Here $D_{vir}$ is the diffusion constant of the extracellular virus and $\gamma_{vir}$ is the decay rate is the decay rate. Uptake and release by a cell at each location are determined using the viral internalization (**E1**) and the viral release (**E3**) modules, and are uniformly applied over all sites of the domain of the cell.

*T2—Cytokine transport*. The change in concentration of the cytokine field $c_{cyt}$ is obtained by solving a reaction-diffusion equation of the following general form,

$$\frac{\partial c_{cyt}}{\partial t} = D_{cyt} \nabla^2 c_{cyt} - \gamma_{cyt} c_{cyt} + s_{cyt}. \tag{21}$$

The decay term $\gamma_{cyt} c_{cyt}$ represents cytokine leaving the simulation domain (*e.g.*, in immune recruitment). To model immune signaling, the rate of cytokine secretion is described by an increasing Hill function of $c_{sig}(\sigma(x))$ with Hill exponent $h_{cyt} = 2$. The meaning of $c_{sig}(\sigma(x))$ depends on the cell type and the Hill exponent can differ for other cell types and states. The rate of cytokine secretion $s_{cyt}$ is written as,

$$s_{cyt}(x, t) = \sigma_{cyt}(\tau(\sigma(x), t)) \frac{(c_{sig}(\sigma(x), t))^{h_{cyt}}}{(c_{sig}(\sigma(x), t))^{h_{cyt}} + (V_{cyt}(\tau(\sigma(x), t)))^{h_{cyt}}} - \omega_{cyt}(\tau(\sigma(x), t)). \tag{22}$$

Here $\sigma_{cyt}(\tau(\sigma(x),t))$ is the maximum cytokine secretion rate for the cell type at $x$, $c_{sig}(\sigma(x))$ is a quantity that affects the cells cytokine secretion, $\omega_{cyt}(\tau(\sigma(x),t))$ is the cytokine uptake rate of the cell type at $x$ and $V_{cyt}(\tau(\sigma(x),t))$ is a dissociation coefficient of cytokine secretion for the cell type at $x$. $\sigma_{cyt}$ is nonzero for infected epithelial cells, virus-releasing epithelial cells and activated immune cells. For infected and virus-releasing epithelial cells $c_{sig}$ is the amount of assembled virus $A$ in the viral replication module, and for activated immune cells $c_{sig}$ is the total amount of cytokine in the domain of the cell. Similarly, for epithelial cells $V_{cyt}$ is the amount of assembled virus, and for immune cells $V_{cyt}$ is the amount of cytokine in the domain of the cell. $\omega_{cyt}(\tau(\sigma(x),t))$ is constant and nonzero for activated and inactive immune cells.

*T3—Oxidizing agent transport*. The oxidizing agent field diffuses according to the reaction-diffusion equation,

$$\frac{\partial c_{oxi}}{\partial t} = D_{oxi} \nabla^2 c_{oxi} - \gamma_{oxi} c_{oxi} + s_{oxi}. \tag{23}$$

Bursts of oxidizing agent are implemented as a source term for one time step according to a rate coefficient $\sigma_{oxi}$, which is uniformly mapped onto the source term $s_{oxi}$ over the domain of

each activated immune cell. An oxidizing burst occurs in immune cells with an activated state when the total cytokine in the immune cell's domain exceeds a threshold $\tau_{oxi}^{sec}$.

**Initial and boundary conditions.** The domain of all simulations had dimensions of 90 x 90 x 2 lattice sites. The initial cell configuration consisted of a 30 x 30 sheet of uninfected epithelial cells, each of size 3 x 3, on the lower layer of lattice sites (see S18 Fig for a demonstration of the negligible effects of a non-uniform arrangement of epithelial cells). Epithelial cells were "frozen", in that they were immobile, leaving the remaining 90 x 90 subdomain for occupancy by recruited immune cells. For cellular dynamics and mass transport, periodic boundary conditions were applied in the plane of the epithelial sheet, and Neumann conditions were applied along the direction orthogonal to the epithelial sheet. All field solutions for the diffusive viral, cytokine and oxidizing agent fields were initialized as zero everywhere.

At each first simulation step, the epithelial cell in the center of the sheet was set to infected, and the amount of internalized virus $U$ of the viral replication model was set to a value of one. All epithelial cells were initialized with a number of unbound surface receptors $SR = R_o$. All immune cells, when introduced to the simulation by recruitment, were initialized in an inactive state, and with a bound cytokine value equal to zero ($B_{cyt} = 0$). During transition of an uninfected epithelial cell to the infected type, all state variables of the viral replication model were initialized with a value of zero.

**Simulation specifications.** Model implementation and all simulations were performed using CompuCell3D, which uses a non-dimensional lattice for CPM-based cellular dynamics and non-dimensional explicit time integration of reaction-diffusion field solutions. As such, a baseline parameter set was constructed for all CPM parameters and modules developed in this work (Table 1). Non-dimensionalization was performed on model parameters for a lattice dimension of 4 μm per pixel along each dimension, at 20 minutes (1/3 hours) per MCS. All replicas were simulated for ten trials, each 1,000 MCS (20000 minutes, 333 hours, 14 days) long. Simulation data was collected at a frequency of 10 MCSs (200 minutes, 3 hours) for all simulations.

## Supporting information

**S1 Fig. No immune system.** Simulation of the progression of infection in a patch of epithelial tissue, with all parameters as in Fig 3, but with no cellular immune system response corresponding to virus spread in an *in vitro* or organoid culture, or a severely immunosuppressed individual. (A) Snapshots of spatial configuration *vs* time showing progression of simulated infection. Columns, left to right: 0 minutes (time of initial infection), 4000 minutes (67 hours, 2 ¾ days) after infection, 8000 minutes (133 hours, 5 ½ days), 12000 minutes (200 hours, 8 ⅓ days), 16000 minutes (267 hours, 11 days), and 20000 minutes (333 hours, 14 days). First row: epithelial cell layer composed of uninfected (blue), infected (green), virus releasing (red) cells and dead cells (black). Second row: level of extracellular virus field. Third row: extracellular cytokine field. Fields are color-coded on a logarithmic scale: red corresponds to the chosen maximum value specified in the first panel, blue to six orders of magnitude lower than the maximum value, and values outside this range are colored as their closest border value. (B-D) Simulation time series. (B) Number of uninfected (orange), infected (green), virus releasing (red) and dead (purple) epithelial cells on a logarithmic scale *vs* time *vs* time in minutes. (C) Total extracellular cytokine (magenta) and total extracellular virus (brown) on a logarithmic scale *vs* time in minutes. (D) Value of the immune recruitment signal $S$ (yellow) and number of immune cells (grey) on a linear scale *vs* time in minutes.
(TIF)

**S2 Fig. Special cases of spatiotemporal infection dynamics.** (A) A border case of slowed infection towards containment is Slowed Infection with constant virus: when the net effectiveness of viral and immune dynamics are balanced, the number of infected cells and the total extracellular virus fluctuate around steady state levels. (B) A limit case of Clearance is the failure to infect: initially infected cells may replicate and secrete virus, but insufficiently so to infect other cells during simulation time such that any initially infected cells vanish and total extracellular virus drops below a threshold of $10^{-3}$ per cell area.
(TIF)

**S3 Fig. Time series of the number of uninfected cells for simulations in Fig 5.** Logarithmic multidimensional parameter sweep performed by running 10 simulation replicas increasing and decreasing the baseline parameter values 10-fold and 100-fold for all parameter sets and replicas in Fig 5. Results show consistent containment/clearance for small $k_{on}$ and small $\beta_{delay}$ (upper right, green-shaded subplots), widespread infection for high $k_{on}$ and small $\beta_{delay}$ (lower left, orange-shaded subplots), and multiple outcomes for the same parameter values (uncolored subplots). Number of cells are shown on a logarithmic scale *vs* time in minutes.
(TIF)

**S4 Fig. Time series of the number of virus releasing cells for simulations in Fig 5.** Logarithmic multidimensional parameter sweep performed by running 10 simulation replicas increasing and decreasing the baseline parameter values 10-fold and 100-fold for all parameter sets and replicas in Fig 5. Results show consistent containment/clearance for small $k_{on}$ and small $\beta_{delay}$ (upper right, green-shaded subplots), widespread infection for high $k_{on}$ and small $\beta_{delay}$ (lower left, orange-shaded subplots), and multiple outcomes for the same parameter values (uncolored subplots). Number of cells are shown on a logarithmic scale *vs* time in minutes.
(TIF)

**S5 Fig. Time series of the number of immune cells for simulations in Fig 5.** Logarithmic multidimensional parameter sweep performed by running 10 simulation replicas increasing and decreasing the baseline parameter values 10-fold and 100-fold for all parameter sets and replicas in Fig 5. Results show consistent containment/clearance for small $k_{on}$ and small $\beta_{delay}$ (upper right, green-shaded subplots), widespread infection for high $k_{on}$ and small $\beta_{delay}$ (lower left, orange-shaded subplots), and multiple outcomes for the same parameter values (uncolored subplots). Number of cells are shown on a logarithmic scale *vs* time in minutes.
(TIF)

**S6 Fig. Time series of the immune response state variable $S$ for simulations in Fig 5.** Logarithmic multidimensional parameter sweep performed by running 10 simulation replicas increasing and decreasing the baseline parameter values 10-fold and 100-fold for all parameter sets and replicas in Fig 5. Results show consistent containment/clearance for small $k_{on}$ and small $\beta_{delay}$ (upper right, green-shaded subplots), widespread infection for high $k_{on}$ and small $\beta_{delay}$ (lower left, orange-shaded subplots), and multiple outcomes for some parameter sets (unshaded subplots). $S$ is shown on a linear scale *vs* time in minutes.
(TIF)

**S7 Fig. Simulations from Fig 8, showing the number of dead cells.** Time series of the number of dead cells for each simulation replica in Fig 8. Number of cells is shown on a logarithmic scale *vs* time in minutes.
(TIF)

**S8 Fig. Simulations from Fig 8, showing the number of infected cells.** Time series of the number of infected cells for each simulation replica in Fig 8. Number of cells is shown on a

logarithmic scale *vs* time in minutes.
(TIF)

**S9 Fig. Simulations from Fig 8 showing the number of immune cells.** Time series of the number of immune cells for each simulation replica in Fig 8. Number of cells is shown on a logarithmic scale *vs* time in minutes.
(TIF)

**S10 Fig. Simulations from Fig 8 showing the immune response state variable.** Time series of the immune response state variable *S* for each simulation replica in Fig 8. *S* is shown on a linear scale *vs* time in minutes.
(TIF)

**S11 Fig. Variation in time of first treatment after infection with a reduced viral RNA replication rate causes a bifurcation in simulation outcomes.** Simulations and parameters are as in Figs 9–11, for a viral replication rate multiplier of 0.375 and, from left to right, time delays of application of 6000, 8000, 10000, and 12000 minutes (100, 133, 167 and 200 hours, 4, 5 ½, 7 and 8 ⅓ days) (dashed lines). Results from all simulation replicas are shown vs time in minutes for, from top to bottom: number of uninfected cells, number of infected cells, number of virus releasing cells, number of dead cells, total extracellular virus, total cytokine, number of immune cells, and immune response state variable *S*. Parameter set subplots are shaded as in Figs 9–11 according to simulation outcomes.
(TIF)

**S12 Fig. Pairwise parameter sweep of the oxidative agent threshold for death $\tau_{oxi}^{death}$ and the virus-receptor association affinity $k_{on}$ (×0.01,× 0.1,× 1,× 10,× 100) around their baseline values, with ten simulation replicas per parameter set (all other parameters have their baseline values as given in Table 1).** The number of uninfected epithelial cells for each simulation replica for each parameter set, plotted on a logarithmic scale, *vs* time displayed in minutes.
(TIF)

**S13 Fig. Pairwise parameter sweep of the immune response delay $\beta_{delay}$ (×0.01,× 0.1,× 1,× 10,× 100) and infection threshold (×0.1,× 0.2,× 1,× 5,× 10) around their baseline values, with ten simulation replicas per parameter set (all other parameters have their baseline values as given in Table 1).** The number of uninfected epithelial cells for each simulation replica for each parameter set, plotted on a logarithmic scale, *vs* time displayed in minutes.
(TIF)

**S14 Fig. Pairwise parameter sweep of the oxidative agent threshold for death $\tau_{oxi}^{death}$ (×0.01,× 0.1,× 1,× 10,× 100) and the infection threshold (×0.1,× 0.2,× 1,× 5,× 10) around their baseline values, with ten simulation replicas per parameter set (all other parameters have their baseline values as given in Table 1).** The number of uninfected epithelial cells for each simulation replica for each parameter set, plotted on a logarithmic scale, *vs* time displayed in minutes.
(TIF)

**S15 Fig. Pairwise parameter sweep of the viral diffusion coefficient $D_{vir}$ and the cytokine diffusion coefficient $D_{cyt}$ (×0.1,× 0.2,× 1,× 5,× 10) around their baseline values, with ten simulation replicas per parameter set (all other parameters have their baseline values as given in Table 1).** The number of uninfected epithelial cells for each simulation replica for

each parameter set, plotted on a logarithmic scale, *vs* time displayed in minutes.
(TIF)

**S16 Fig. Pairwise parameter sweep of the oxidative agent threshold for death $\tau_{oxi}^{death}$ and the virally-induced apoptosis dissociation coefficient $V_{apo}$ (×0.01, ×0.1, ×1, ×10, ×100) around their baseline values, with ten simulation replicas per parameter set (all other parameters have their baseline values as given in Table 1).** The number of uninfected epithelial cells for each simulation replica for each parameter set, plotted on a logarithmic scale, *vs* time displayed in minutes.
(TIF)

**S17 Fig. Pairwise parameter sweep of the viral diffusion coefficient $D_{vir}$ (×0.1, ×0.2, ×1, ×5, ×10) and the virus-receptor association affinity $k_{on}$ (×0.01, ×0.1, ×1, ×10, ×100) around their baseline values, with ten simulation replicas per parameter set (all other parameters have their baseline values as given in Table 1).** The number of uninfected epithelial cells for each simulation replica for each parameter set, plotted on a logarithmic scale, *vs* time displayed in minutes.
(TIF)

**S18 Fig. Simulation of viral infection using the baseline parameter set as in Fig 3 but with a non-uniform epithelial sheet.** (A) Widespread infection occurs with the same spatiotemporal features as in Fig 3A in a non-uniform epithelial sheet. (B) Ten simulation replicas with a non-uniform epithelial sheet showed no significant differences in transient metrics compared to simulations with a uniform epithelial sheet.
(TIF)

**S1 Table. Varying parameters in simulations shown in Fig 4.** Virus-receptor association affinity and immune response delay coefficient shown for no immune response (Fig 4A), widespread infection (Fig 4B), slowed infection (Fig 4C), containment (Fig 4D), recurrence (Fig 4E) and clearance (Fig 4F). All other parameters are as in Table 1.
(DOCX)

**S2 Table. Parameter values of integrated HCV model.**
(DOCX)

**S1 Code Snippet. Select import and load commands from Simulation/ViralInfectionVTM.py.**
(TIF)

**S2 Code Snippet. API for the steppable implementing the Simple Recovery model, derived from Models/RecoverySimple/RecoverySteppables.py.** The exact code of the implementation is shown for the steppable function "cell_recovers" (Lines 32–36).
(TIF)

**S3 Code Snippet. API for the steppable implementing the Simple Recovery model, derived from Models/RecoveryNeighbor/RecoverySteppables.py.**
(TIF)

**S1 Text. Integration of an explicit RNA synthesis model.**
(DOCX)

**S2 Text. Collaborative viral infection modeling environment.**
(DOCX)

## Acknowledgments

We would like to thank Ramray Bhat, Ryan Danehy, Javier Toledo, Bard Ermentrout, Geoffrey Fox, Yi Jiang, Paul Macklin, Jairaj Mathur, Roeland Merks, Ericka Mochan, Jason Shoemaker and Amber Smith for useful discussions and ideas and for comments on the manuscript.

## Author Contributions

**Conceptualization:** T. J. Sego, Josua O. Aponte-Serrano, Juliano Ferrari Gianlupi, Lutz Brusch, James M. Osborne, James A. Glazier.

**Data curation:** T. J. Sego, Josua O. Aponte-Serrano, Juliano Ferrari Gianlupi.

**Formal analysis:** T. J. Sego, Josua O. Aponte-Serrano, Juliano Ferrari Gianlupi, Kira Breithaupt, Jessica Crawshaw.

**Funding acquisition:** James A. Glazier.

**Investigation:** T. J. Sego, Josua O. Aponte-Serrano, Juliano Ferrari Gianlupi, Samuel R. Heaps, Kira Breithaupt, Jessica Crawshaw, Ellen M. Quardokus.

**Methodology:** T. J. Sego, Josua O. Aponte-Serrano, Juliano Ferrari Gianlupi, Kira Breithaupt, Lutz Brusch, James M. Osborne, James A. Glazier.

**Project administration:** T. J. Sego, James A. Glazier.

**Resources:** T. J. Sego, James A. Glazier.

**Software:** T. J. Sego, Josua O. Aponte-Serrano, Juliano Ferrari Gianlupi, Kira Breithaupt, James A. Glazier.

**Supervision:** James A. Glazier.

**Validation:** T. J. Sego, Josua O. Aponte-Serrano, Juliano Ferrari Gianlupi, Lutz Brusch, James M. Osborne, James A. Glazier.

**Visualization:** T. J. Sego, Josua O. Aponte-Serrano, Ellen M. Quardokus.

**Writing – original draft:** T. J. Sego, Josua O. Aponte-Serrano, Juliano Ferrari Gianlupi, Samuel R. Heaps, Kira Breithaupt, Lutz Brusch, James M. Osborne, Ellen M. Quardokus, James A. Glazier.

**Writing – review & editing:** T. J. Sego, Josua O. Aponte-Serrano, Juliano Ferrari Gianlupi, Samuel R. Heaps, Kira Breithaupt, Lutz Brusch, Jessica Crawshaw, James M. Osborne, Ellen M. Quardokus, Richard K. Plemper, James A. Glazier.

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
