## [Decision Letter · Decision Letter 0]

20 Jul 2020

Dear Mr Aponte-Serrano,

Thank you very much for submitting your manuscript "A modular framework for multiscale, multicellular, spatiotemporal modeling of acute primary viral infection and immune response in epithelial tissues and its application to drug therapy timing and effectiveness" for consideration at PLOS Computational Biology.

As with all papers reviewed by the journal, your manuscript was reviewed by members of the editorial board and by several independent reviewers. In light of the reviews (below this email), we would like to invite the resubmission of a significantly-revised version that takes into account the reviewers' comments.

This is a timely manuscript that presents an interesting model which can be extended in the future by this team and by others. The reviewers were generally enthusiastic, but they have important specific suggestions and critiques that should be addressed to improve the paper.

We cannot make any decision about publication until we have seen the revised manuscript and your response to the reviewers' comments. Your revised manuscript is also likely to be sent to reviewers for further evaluation.

Sincerely,

Shayn M Peirce

Guest Editor

PLOS Computational Biology

Feilim Mac Gabhann

Editor-in-Chief

PLOS Computational Biology

This is a timely manuscript that presents an interesting model which can be extended in the future by this team and by others. The reviewers were generally enthusiastic, but they have important specific suggestions and critiques that should be addressed to improve the paper.

Reviewer's Responses to Questions

**Comments to the Authors:**

Reviewer #1: In this paper the authors address the important issue of viral - immune system interactions within epithelial tissues. The new modelling presented in this paper is distributed per three components, namely: epithelium, ECM, and lymph node component. The open source modelling framework proposed here aims to explore computationally the outcomes of potential treatments.

The novel modelling and computational framework presented in this paper brings a significant contribution towards advancing the understanding of viral infection, and as a consequence I recommend this paper to be accepted for immediate publication in PLOS Computational Biology.

Below, I have a small list of questions and a suggestion regarding the paper structure.

Questions:

Q1: Would it be possible to add more details regarding the way the ECM is accounted for within the model? I am not asking to modify the model, but I feel that the reader would benefit if a paragraph would be added under the Model and Methods section to clarify how the ECM is implicitly/explicitly considered within the model.

Q2: Could the authors comment on how would a heterotypic population of tissue cells make a difference to the modelling (that could perhaps be considered in a future work) and give a hint on what would be a potential impact for the computational results?

Q3: Could the authors add more comments on the perspective of considering more than one generic immune cell type (as this is a heterotypic system) and give a hint on how would that be beneficial to the investigation?

Finally, as a suggestion regarding the paper structure and presentation, it would help the reader if either the entire section Models and Methods (or at least a brief overview of this) would be relocated/placed in the manuscript before the Results section.

Reviewer #2: Sego et al present a computational model of viral infection in an epithelial sheet that includes some aspects of the innate immune response. The model seems to be a reasonable abstraction and qualitatively matches the temporal course of viral infections. As the authors note, its presentation here is to provide an open-source framework for future studies, and as such, it is a useful contribution. In the current manuscript, the key finding cited by the authors is that low levels of inhibition of viral replication, e.g., by a low dose of a drug, can improve infection outcomes more if administered earlier in the progression of viral infection. That is not a surprising finding, but is a reasonable check on the model’s reasonableness. Frankly, I do worry that without a more quantitative alignment with experiments/observations, that the model will not find any more substantial use in understanding infection course or treatment options.

Some specific concerns

1. The introduction provides a detailed overview of viral infections, including a paragraph on the adaptive immune response that is not actually part of the model. In so doing, the introduction somewhat overpromises in the sense that the model presented later omits many of these details and lumps together others. I would advise the authors to revise the introduction to make clear early on that while the authors are aware of the complexity of the immune response, the model’s abstraction of the immune response will be greatly simplified.

2. Lines 215-217. The authors note that they “established a baseline set of parameters (Table 1) for which the immune response is strong enough to slow the spread of infection, but insufficient to prevent widespread infection and death of all epithelial cells (Figure 3).” That is a reasonable starting point and the authors later go on to vary some of these parameters by orders of magnitude to map out model time courses across parts of parameter space, but is this baseline parameter set in any sense unique? For example, looking at Figure 5, the results for the baseline parameters (multipliers of 1 and 1) are in that intermediate range, but so are the results for multipliers of 100 and 0.1 (or 0.1 and 100). Why was the {1,1}-multiplier parameter set chosen as baseline rather than one of the other combinations that yielded similar results? Are there other parameter pairs (or triplets) that were not subjected to such large parameter sweeps that could compensate for one another? The authors need to provide more detail/justification for their choice of baseline parameter set.

3. Given the banded nature of the results in Figs 5-10, the complex model constructed by the authors seems to reduce to a comparison of two rates: one related to viral replication and spread; the other related to cell death (both viral and immune-mediated). The authors could increase the paper’s impact by using their model to explore how its parameters impact the relevant rates and the parameter-space locations of the transitions where one rate dominates the other. Helping the community build such intuition would be an excellent use of the model and would increase the likelihood that others would be able to use it effectively in future work.

4. Lines 677-679. The authors state that “While the simulations presented here begin with a single infected cell, a simulation which begins with multiple infected cells might better represent the infection dynamics of patients that have been subject to high level exposure …” It seems to me that the model may already be simulating a fairly high-level exposure. Since it uses periodic boundary conditions for both the cells and the diffusing fields, I don’t think the model is simulating a single infected cell in a large epithelium, but rather is simulating a case where 1 in 900 cells is infected (ad infinitum repetition of this 900-cell field is implied by the periodic boundary conditions). The authors should explore the literature on viral infections of epithelia and evaluate whether an initial infection of 1 in 900 cells is really a low- or high-level exposure.

5. I appreciate that the authors are making the code for their model open-source, but the “Models and methods” section should still be detailed enough for others to reproduce the model independently. I found the separate presentation of the Conceptual and Quantitative aspects of the model a bit difficult to follow, but that is somewhat a matter of taste. Within the Quantitative sections, I have the following questions / concerns:

a. Under E2 – Viral Replication, the description of U as “unpacking” is confusing. “Unpacking” is a process, so it is confusing as to whether U is the amount of virus that has been unpacked or the amount that has not yet been unpacked. Given the structure of Eqn 6, I suspect it is the latter, since dU/dt increases as Uptake increases and decreases in proportion to the amount of U currently in existence. This should be clarified as should all the descriptions in that section. Each of the variables U, R, P and A should be explicitly described as the amount of some object/molecule rather than as a process. The time derivatives are the process.

b. Line 945. Isn’t the replication rate reduced to (rmax/2)R rather than just rmax/2.

c. Under E4 – Cell death. What happens to internal assembled virus upon virus- or immune-induced cell death? Are these assembled viruses released or destroyed in the model?

d. Why were all the Hill factors chosen to be 2? Why is a Hill factor not listed as a parameter in Eqn 21 and is instead simply designated as 2?

e. The authors should carefully check all their del and gradient symbols. Some are inverted from the others and it is not clear that this is intentional.

f. Section I3 – Immune cell direct cytotoxicity and bystander effect. This short section is very unclear. I cannot tell what is actually coded.

g. Section I4 is missing any math details.

h. Section T2- Cytokine transport. Eqn 21 is presented with a very general c_sig that is only described as “a quantity that affects its cytokine secretion.” This initially left me very confused until later in the paragraph where c_sig is noted to be different quantities in different situations (amount of assembled virus or local cytokine concentration). Given that these are two very different means of triggering cytokine secretion, why do they both have the same dissociation coefficient V_cyt?

i. In the table of baseline parameter values (Table 1), the formatting of some entries is ambiguous, e.g., is 1/6x10^(-3) really 0.167 x 10^(-3) or 1/(6x10^(-3))? And towards the end of the table, there are several entries where items that should be superscripts are not.

Reviewer #3: In this article, the authors present a multiscale, multicellular computational framework to model the spatio-temporal dynamics of viral spread and immune responses during acute infection. Analyzing different scenarios they use their model to investigate the effect of different drug therapy regimens on disease outcome.

The paper is well written and the structure of the modeling framework is explained. However, in my view, the presented results do not provide novel insights for understanding viral infections or drug therapy in general or for specific viruses, nor do they emphasize the necessity to consider spatio-temporal dynamics when analyzing viral spread. Using the model to study specific viral infections would also require careful parameterizations and adaption of a lot of aspects of this model to this particular virus so that it is not clear how this would provide a general framework for analyzing viral spread in its current stage.

# Major points:

(1.) The simulation environment is quite complex including a lot of different factors and processes to capture relevant aspects during acute infection. In order to get meaningful results and insights, parameterization of these processes according to experimental or clinical data is the key, and the most difficult part. For most of the parameters, and especially the virus related ones, no sources for the choice of parameters are given (Table 1). It is not clear if the model was parameterized considering a specific virus but the later statements (line 657-658) indicate that this is only work in progress. It is questionable how meaningful the results are if arbitrary values were chosen, even if they were used for qualitative comparisons.

(2.) Although modeling spatio-temporal dynamics of viral spread, the analysis is done on a simple squared lattice of epithelial cells. To make an argument for considering spatial aspects, the authors need to investigate how much the results would differ if they would use population-based ODE-models instead (as they acknowledge themselves, line 598-604). Without investigating processes that specifically imply considering spatial aspects (tissue regeneration, cell-transmission modes, local immune responses) the complexity of the model would not necessarily provide additional insights.

(3.) There have been a number of reviews that address the multi-scale modeling of viral spread and immune responses, and when it is necessary to consider spatial aspects (e.g. Bauer et al. Inf Sciences 2009, Graw & Perelson Annals of Virology 2016). They cite several studies that have used spatially-resolved multi-scale models for different viruses.

(4.) Many viruses (HIV, HCV, Influenza A, ..) have different replication mechanisms, target tissues and relevant immune responses. That means, despite providing a modeling framework, many aspects of the model need to be adapted to investigate the spatio-temporal dynamics for a particular virus. It is not clear how flexible the modeling framework is for this task.

(5.) The aspects of time of treatment and drug effectiveness have been investigated using ODE-models, even for SARS-CoV-2. The studies come to similar conclusions without the need to consider spatial aspects and by analyzing clinical data, i.e., obtaining relevant parameterizations (Goncalves et al. CPT Pharmacometrics 2020, Goyal et al. medRxiv 2020).

# Minor points:

- The model and the analysis itself (tissue structure, parameterization, pharmacodynamics) is not COVID-specific. Therefore, using COVID-19 as a keyword or prominent “example” is misleading.

**Have all data underlying the figures and results presented in the manuscript been provided?**

Reviewer #1: Yes

Reviewer #2: Yes

Reviewer #3: None

PLOS authors have the option to publish the peer review history of their article (what does this mean?). If published, this will include your full peer review and any attached files.

Reviewer #1: No

Reviewer #2: No

Reviewer #3: No
---

## [Decision Letter · Decision Letter 1]

20 Oct 2020

Dear Dr Sego,

We are pleased to inform you that your manuscript 'A modular framework for multiscale, multicellular, spatiotemporal modeling of acute primary viral infection and immune response in epithelial tissues and its application to drug therapy timing and effectiveness' has been provisionally accepted for publication in PLOS Computational Biology.

Best regards,

Shayn M Peirce

Guest Editor

PLOS Computational Biology

Feilim Mac Gabhann

Editor-in-Chief

PLOS Computational Biology

Reviewer's Responses to Questions

**Comments to the Authors:**

Reviewer #1: The authors have addressed all the points I raised in my review, and so I recommend the paper to be accepted in the current form for immediate publication.

Reviewer #2: The authors have largely addressed my previous concerns. However, they have added a new 8-page section to Results entitled "Model extensions" that includes preliminary investigations. These are frankly not as complete as the rest of the results. I actually feel that these sections distract from the paper's primary focus, providing more of a survey than deep insight into particular questions. They make an already long paper a bit more unwieldy.

**Have all data underlying the figures and results presented in the manuscript been provided?**

Reviewer #1: Yes

Reviewer #2: Yes

PLOS authors have the option to publish the peer review history of their article (what does this mean?). If published, this will include your full peer review and any attached files.

Reviewer #1: No

Reviewer #2: No

---

## [Editor Report · Acceptance letter]

28 Nov 2020

PCOMPBIOL-D-20-00929R1 

A modular framework for multiscale, multicellular, spatiotemporal modeling of acute primary viral infection and immune response in epithelial tissues and its application to drug therapy timing and effectiveness

Dear Dr Sego,

I am pleased to inform you that your manuscript has been formally accepted for publication in PLOS Computational Biology. Your manuscript is now with our production department and you will be notified of the publication date in due course.

With kind regards,

Nicola Davies
